# Improved Sample Complexity Bounds for Diffusion Model Training

**Shivam Gupta**
UT Austin
shivamgupta@utexas.edu

**Aditya Parulekar**
UT Austin
adityaup@cs.utexas.edu

**Eric Price**
UT Austin
ecprice@cs.utexas.edu

**Zhiyang Xun**
UT Austin
zxun@cs.utexas.edu

## Abstract

Diffusion models have become the most popular approach to deep generative modeling of images, largely due to their empirical performance and reliability. From a theoretical standpoint, a number of recent works [CCL+23b, CCSW22, BBDD24] have studied the iteration complexity of sampling, assuming access to an accurate diffusion model. In this work, we focus on understanding the *sample complexity* of training such a model; how many samples are needed to learn an accurate diffusion model using a sufficiently expressive neural network? Prior work [BMR20] showed bounds polynomial in the dimension, desired Total Variation error, and Wasserstein error. We show an *exponential improvement* in the dependence on Wasserstein error and depth, along with improved dependencies on other relevant parameters.

## 1   Introduction

Score-based diffusion models are currently the most successful methods for image generation, serving as the backbone for popular text-to-image models such as stable diffusion [RBL+22], Midjourney, and DALL·E [RDN+22] as well as achieving state-of-the-art performance on other audio and image generation tasks [SDWMG15, HJA20, JAD+21, SSXE22, DN21].

The goal of score-based diffusion is to produce a generative model for a possibly complicated distribution $q_0$. This involves two components: *training* a neural network using true samples from $q_0$ to learn estimates of its (smoothed) score functions, and *sampling* using the trained estimates. To this end, consider the following stochastic differential equation, often referred to as the *forward* SDE:

$$\mathrm{d}x_t = -x_t\,\mathrm{d}t + \sqrt{2}\,\mathrm{d}B_t, \quad x_0 \sim q_0 \tag{1}$$

where $B_t$ represents Brownian motion. Here, $x_0$ is a sample from the original distribution $q_0$ over $\mathbb{R}^d$, while the distribution of $x_t$ can be computed to be

$$x_t \sim e^{-t}x_0 + \mathcal{N}(0, \sigma_t^2 I_d)$$

for $\sigma_t^2 = 1 - e^{-2t}$. Note that this distribution approaches $\mathcal{N}(0, I_d)$, the stationary distribution of (1), exponentially fast.

Let $q_t$ be the distribution of $x_t$, and let $s_t(y) := \nabla \log q_t(y)$ be the associated *score* function. We refer to $q_t$ as the $\sigma_t$-*smoothed* version of $q_0$. Then, starting from a sample $x_T \sim q_T$, there is a reverse

---

Authors listed in alphabetical order.

SDE associated with the above forward SDE in (1) [And82]:

$$\mathrm{d}x_{T-t} = (x_{T-t} + 2s_{T-t}(x_{T-t}))\,\mathrm{d}t + \sqrt{2}\,\mathrm{d}B_t. \tag{2}$$

That is to say, if we begin at a sample $x_T \sim q_T$, following the reverse SDE in (2) back to time 0 will give us a sample from the *original* distribution $q_0$. This suggests a natural strategy to sample from $q_0$: start at a large enough time $T$ and follow the reverse SDE back to time 0. Since $x_T$ approaches $\mathcal{N}(0, I_d)$ exponentially fast in $T$, our samples at time 0 will be distributed close to $q_0$. In particular, if $T$ is large enough—logarithmic in $\frac{m_2}{\varepsilon}$—then samples produced by this process will be $\varepsilon$-close in TV to being drawn from $q_0$. Here $m_2^2$ is the second moment of $q_0$, given by $m_2^2 := \mathbb{E}_{x \sim q_0}[\|x\|^2]$.

In practice, this continuous reverse SDE (2) is approximated by a time-discretized process. That is, the score $s_{T-t}$ is approximated at some fixed times $0 = t_0 \leq t_1 \leq \cdots \leq t_k < T$, and the reverse process is run using this discretization, holding the score term constant at each time $t_i$. This algorithm is referred to as "DDPM", as defined in [HJA20].

Chen et al. [CCL⁺23b] proved that as long as we have access to sufficiently accurate estimates for each score $s_t$ at each discretized time, this reverse process produces accurate samples. Specifically, for *any* $d$-dimensional distribution $q_0$ supported on the Euclidean Ball of radius $R$, the reverse process can sample from a distribution $\varepsilon$-close in TV to a distribution $(\gamma \cdot R)$-close in 2-Wasserstein to $q_0$ in $\mathrm{poly}(d, 1/\varepsilon, 1/\gamma)$ steps, as long as the score estimates $\widehat{s}_t$ used at each step are $\widetilde{O}(\varepsilon^2)$ accurate in squared $L^2$. That is, as long as

$$\mathbb{E}_{x \sim q_t}[\|\widehat{s}_t(x) - s_t(x)\|^2] \leq \widetilde{O}(\varepsilon^2). \tag{3}$$

In this work, we focus on understanding the *sample complexity* of learning such score estimates. Specifically, we ask:

> How many samples are required for a sufficiently expressive neural network to learn accurate score estimates that generate high-quality samples using the DDPM algorithm?

We consider a training process that employs the empirical minimizer of the *score matching objective* over a class of neural networks as the score estimate. More formally, we consider the following setting:

**Setting 1.1.** *Let $\mathcal{F}(D, P, \Theta)$ be the class of functions represented by a fully connected neural network with ReLU activations and depth $D$, with $P$ parameters, each bounded by $\Theta$. Let $\{t_k\}$ be some time discretization of $[0, T]$. Given $m$ i.i.d. samples $x_i \sim q_0$, for each $t \in \{t_k\}$, we take $m$ Gaussian samples $z_i \sim \mathcal{N}(0, \sigma_t^2 I_d)$. We take $\widehat{s}_t$ to be the minimizer of the score-matching objective:*

$$\widehat{s}_t = \arg\min_{f \in \mathcal{F}} \frac{1}{m} \sum_{i=1}^{m} \left\| f\left(e^{-t}x_i + z_i\right) - \frac{-z_i}{\sigma_t^2} \right\|_2^2. \tag{4}$$

*We then use $\{\widehat{s}_{t_k}\}$ as the score estimates in the DDPM algorithm.*

This is the same setting as is used in practice, except that in practice (4) is optimized with SGD rather than globally. As in [CCL⁺23b], we aim to output samples from a distribution that is $\varepsilon$-close in TV to a distribution that is $\gamma R$ or $\gamma m_2$-close in Wasserstein to $q_0$. We thus seek to bound the number of samples $m$ to get such a good output distribution, in terms of the parameters of Setting 1.1 and $\varepsilon, \gamma$.

Block et al. [BMR20] first studied the sample complexity of learning score estimates using the empirical minimizer of the score-matching objective (4). They showed sample complexity bounds that depend on the Rademacher complexity of $\mathcal{F}$. Applied to our setting and using known bounds on Rademacher complexity of neural networks, in Setting 1.1 their result implies a sample complexity bound of $\widetilde{O}\left(\frac{d^{5/2}}{\gamma^3 \varepsilon^2}(\Theta^2 P)^D \sqrt{D}\right)$. See Appendix E for a detailed discussion.

Following the analysis of DDPM by [CCL⁺23b], more recent work on the iteration complexity of *sampling* [CLL22, BBDD24] has given an *exponential improvement* on the Wasserstein accuracy $\gamma$, as well as replacing the uniform bound $R$ by $m_2$ — the square root of the second moment. In

---

$\widetilde{O}$ hides polylogarithmic factors in $d$, $\frac{1}{\varepsilon}$ and $\frac{1}{\gamma}$

particular, [BBDD24] show that $\widetilde{O}(\frac{d}{\varepsilon^2} \log^2 \frac{1}{\gamma})$ iterations suffice to sample from a distribution that is $\gamma \cdot m_2$ close to $q_0$ in 2-Wasserstein, as long as the score estimates are $\widetilde{O}\left(\varepsilon^2/\sigma_t^2\right)$ accurate, i.e.,

$$\mathop{\mathbb{E}}_{x \sim q_t}[\|\widehat{s}_t(x) - s_t(x)\|^2] \leq \widetilde{O}\left(\frac{\varepsilon^2}{\sigma_t^2}\right). \tag{5}$$

Inspired by these works, we ask: is it possible to achieve a similar exponential improvement in the sample complexity of *learning* score estimates using a neural network?

## 1.1 Our Results

We give a new sample complexity bound of $\widetilde{O}(\frac{d^2}{\varepsilon^3} PD \log \Theta \log^3 \frac{1}{\gamma})$ for learning scores to sufficient accuracy for sampling via DDPM. Learning is done by optimizing the same score matching objective that is used in practice. Compared to [BMR20], our bound has *exponentially* better dependence on $\Theta, \gamma$, and $D$, and a better polynomial dependence on $d$ and $P$, at the cost of a worse polynomial dependence in $\varepsilon$.

As discussed above, for the sampling process, it suffices for the score estimate $s_t$ at time $t$ to have error $\widetilde{O}\left(\varepsilon^2/\sigma_t^2\right)$. This means that scores at larger times need higher accuracy, but they are also intuitively easier to estimate because the corresponding distribution is smoother. Our observation is that the two effects cancel out: we show that the sample complexity to achieve this accuracy for a fixed $t$ is *independent* of $\sigma_t$. However, this only holds once we weaken the accuracy guarantee slightly (from $L^2$ error to "$L^2$ error over a $1 - \delta$ fraction of the mass"). We show that this weaker guarantee is nevertheless sufficient to enable accurate sampling. Our approach lets us run the SDE to a very small final $\sigma_t = \gamma$, which yields a final $\gamma$ dependence of $O(\log^3 \frac{1}{\gamma})$ via a union bound over the times $t$ that we need score estimates $s_t$ for. In contrast, the approach in [BMR20] gets the stronger $L^2$-accuracy, but requires a $\mathrm{poly}(\frac{1}{\sigma_t})$ dependence in sample complexity for each score $s_t$; this leads to their $\mathrm{poly}(\frac{1}{\gamma})$ sample complexity overall.

To state our results formally, we make the following assumptions on the data distribution and the training process:

**A1** The second moment $m_2^2$ of $q_0$ is between $1/\mathrm{poly}(d)$ and $\mathrm{poly}(d)$.

**A2** For the score $s_t$ used at each step, there exists some function $f \in \mathcal{F}(D, P, \Theta)$ (as defined in Setting 1.1) such that the $L^2$ error, $\mathbb{E}_{x \sim q_t}[\|f(x) - s_t(x)\|^2]$, is sufficiently small.

That is: the data is somewhat normalized, and the smoothed scores can be represented well in the function class. Our main theorem is as follows:

**Theorem 1.2.** *In Setting 1.1, suppose assumptions A1 and A2 hold. For any $\gamma > 0$, consider the score functions trained from*

$$m \geq \widetilde{O}\left(\frac{d^2 PD}{\varepsilon^3} \cdot \log \Theta \cdot \log^3 \frac{1}{\gamma}\right)$$

*i.i.d. samples of $q_0$. With 99% probability, DDPM using these score functions can sample from a distribution $\varepsilon$-close in $\mathsf{TV}$ to a distribution $\gamma m_2$-close to $q$ in 2-Wasserstein.*

We remark that assumption A1 is made for a simpler presentation of our theorem; the (logarithmic) dependence on the second moment is analyzed explicitly in Theorem C.2. The quantitative bound for A2—exactly how small the $L^2$ error needs to be—is given in detail in Theorem C.3.

**Barrier for $L^2$ accuracy.** As mentioned before, previous works have been using $L^2$ accurate score estimation either as the assumption for sampling or the goal for training. Ideally, one would like to simply show that the ERM of the score matching objective will have bounded $L^2$ error of $\varepsilon^2/\sigma^2$ with a number of samples that scales polylogarithmically in $\frac{1}{\sigma}$. Unfortunately, this is *false*. In fact, it is information-theoretically impossible to achieve this in general without $\mathrm{poly}(\frac{1}{\sigma})$ samples. Since sampling to $\gamma m_2$ Wasserstein error needs to consider a final $\sigma_t = \gamma$, this leads to a $\mathrm{poly}(\frac{1}{\gamma})$ dependence. See Figure 1, or the discussion in Section 4, for a hard instance.

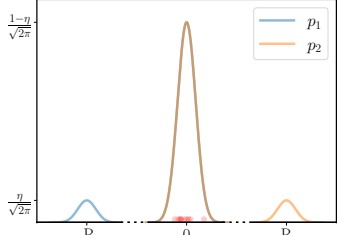 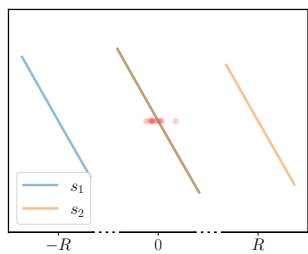 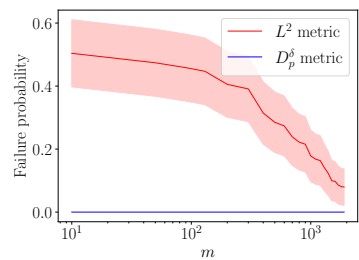

Figure 1: Given $o\left(\frac{1}{\eta}\right)$ samples from either $p_1 = (1-\eta)\mathcal{N}(0,1) + \eta\mathcal{N}(-R,1)$, or $p_2 = (1-\eta)\mathcal{N}(0,1) + \eta\mathcal{N}(R,1)$ we will only see samples from the main Gaussian with high probability, and cannot distinguish between them. However, if we pick the wrong score function, the $L^2$ error incurred is large - about $\eta R^2$. On the right, we take $\eta = 0.001, R = 10000, \delta = 0.01$. We plot the probability that the ERM has error larger than 0 in the $L^2$ sense, and our $D_p^\delta$ sense.

In the example in Figure 1, score matching + DDPM still *works* to sample from the distribution with sample complexity scaling with $\text{poly}(\log\frac{1}{\gamma})$; the problem lies in the theoretical justification for it. Given that it is impossible to learn the score in $L^2$ to sufficient accuracy with fewer than $\text{poly}(\frac{1}{\gamma})$ samples, such a justification needs a different measure of estimation error. We will introduce such a measure, showing (1) that it will be small for all relevant times $t$ after a number of samples that scales *polylogarithmically* in $\frac{1}{\gamma}$, and (2) that this measure suffices for fast sampling via the reverse SDE.

The problem with measuring error in $L^2$ comes from outliers: rare, large errors can increase the $L^2$ error while not being observed on the training set. We proved that we can relax the $L^2$ accuracy requirement for diffusion model to the $(1-\delta)$-quantile error: For distribution $p$, and functions $f, g$, we say that

$$D_p^\delta(f,g) \leq \varepsilon \iff \mathbb{P}_{x\sim p}\left[\|f(x) - g(x)\|_2 \geq \varepsilon\right] \leq \delta. \tag{6}$$

We adapt [BBDD24] to show that learning the score in our new outlier-robust sense also suffices for sampling, if we have accurate score estimates at each relevant discretization time.

**Lemma 1.3.** *Let $q$ be a distribution over $\mathbb{R}^d$ with second moment $m_2^2$ between $1/poly(d)$ and $poly(d)$. For any $\gamma > 0$, there exist $N = \widetilde{O}\left(\frac{d}{\varepsilon^2+\delta^2}\log^2\frac{1}{\gamma}\right)$ discretization times $0 = t_0 < t_1 < \cdots < t_N < T$ such that if the following holds for every $k \in \{0,\ldots,N-1\}$:*

$$D_{q_{T-t_k}}^{\delta/N}\left(\widehat{s}_{T-t_k}, s_{T-t_k}\right) \leq \frac{\varepsilon}{\sigma_{T-t_k}}$$

*then DDPM can sample from a distribution that is within $\widetilde{O}(\delta + \varepsilon\sqrt{\log(d/\gamma)})$ in TV distance to $q_\gamma$ in $N$ steps.*

With this relaxed requirement, our main technical lemma shows that for each fixed time, a good $(1-\delta)$-quantile accuracy can be achieved with a small number of samples, independent of $\sigma_t$.

**Lemma 1.4** (Main Lemma). *In Setting 1.1, suppose assumptions **A1** and **A2** hold. By taking $m$ i.i.d. samples from $q_0$ to train score function $\widehat{s}_t$, when*

$$m > \widetilde{O}\left(\frac{(d + \log\frac{1}{\delta_{train}}) \cdot PD}{\varepsilon^2 \delta_{score}} \cdot \log\left(\frac{\Theta}{\delta_{train}}\right)\right),$$

*with probability $1 - \delta_{train}$, the score estimate $\widehat{s}_t$ satisfies*

$$D_{q_t}^{\delta_{score}}(\widehat{s}_t, s_t) \leq \varepsilon/\sigma_t.$$

Combining Lemma 1.4 with Lemma 1.3 gives us Theorem 1.2. The proof is detailed in Appendix C.

## 2 Related Work

Score-based diffusion models were first introduced in [SDWMG15] as a way to tractably sample from complex distributions using deep learning. Since then, many empirically validated techniques

have been developed to improve the sample quality and performance of diffusion models [HJA20, ND21, SE20, SSDK+21, SDME21]. More recently, diffusion models have found several exciting applications, including medical imaging and compressed sensing [JAD+21, SSXE22], and text-to-image models like DALL·E 2 [RDN+22] and Stable Diffusion [RBL+22].

Recently, a number of works have begun to develop a theoretical understanding of diffusion. Different aspects have been studied – the sample complexity of training with the score-matching objective [BMR20], the number of steps needed to sample given accurate scores [CCL+23b, CLL22, CCSW22, CCL+23a, BBDD24, BTHD21, LLT23], and the relationship to more traditional methods such as maximum likelihood [PRS+23, KHR23].

On the *training* side, [BMR20] showed that for distributions bounded by $R$, the score-matching objective learns the score of $q_\gamma$ in $L^2$ using a number of samples that scales polynomially in $\frac{1}{\gamma}$. On the other hand, for *sampling* using the reverse SDE in (2), [CLL22, BBDD24] showed that the number of steps to sample from $q_\gamma$ scales polylogarithmically in $\frac{1}{\gamma}$ given $L^2$ approximations to the scores.

Our main contribution is to show that while learning the score in $L^2$ *requires* a number of samples that scales polynomially in $\frac{1}{\gamma}$, the score-matching objective *does learn* the score in a weaker sense with sample complexity depending only *polylogarithmically* in $\frac{1}{\gamma}$. Moreover, this weaker guarantee is sufficient to maintain the polylogarithmic dependence on $\frac{1}{\gamma}$ on the number of steps to sample with $\gamma \cdot m_2$ 2-Wasserstein error.

Our work, as well as [BMR20], assumes the score can be accurately represented by a small function class such as neural networks. Another line of work examines what is possible for more general distributions [CHZW23, OAS23]. For example, [CHZW23] shows that for general subgaussian distributions, the scores can be learned to $L^2$ error $\varepsilon/\sigma_t$ with $(d/\varepsilon)^{O(d)}$ samples. Our approach avoids this exponential dependence, but assumes that neural networks can represent the score.

## 3 Proof Overview

In this section, we outline the proofs for two key lemmas: Lemma 1.4 in Section 3.1 and Lemma 1.3 in Section 3.2. The complete proofs for these lemmas are provided in Appendix A and Appendix B respectively.

### 3.1 Training

We show that the score-matching objective (4) concentrates well enough that the ERM is close to the true minimizer. Prior work on sampling [BBDD24] shows that estimating the $\sigma$-smoothed score to $L_2^2$ error of $\frac{\varepsilon^2}{\sigma^2}$ suffices for sampling; our goal is to get something close to this with a sample complexity independent of $\sigma$.

**Background: Minimizing the true expectation gives the true score.** In this section, we show that if we could compute the true expectation of the score matching objective, instead of just the empirical expectation, then the true score would be its minimizer. For a fixed $t$, let $\sigma = \sigma_t$ and $p$ be the distribution of $e^{-t}x$ for $x \sim q_0$. We can think of a joint distribution of $(y, x, z)$ where $y \sim p$ and $z \sim N(0, \sigma^2 I_d)$ are independent, and $x = y + z$ is drawn according to $q_t$. With this change of variables, the score matching objective used in (4) is

$$\mathbb{E}_{x,z}\left[\left\|s(x) - \frac{-z}{\sigma^2}\right\|_2^2\right].$$

Because $x = y + z$ for Gaussian $z$, Tweedie's formula states that the true score $s^* = s_t$ is given by

$$s^*(x) = \mathbb{E}_{z|x}\left[\frac{-z}{\sigma^2}\right].$$

Define $\Delta = s^*(x) - \frac{-z}{\sigma^2}$, so $\mathbb{E}[\Delta \mid x] = 0$. Therefore for any $x$,

$$l(s, x, z) := \left\|s(x) - \frac{-z}{\sigma^2}\right\|_2^2 \tag{7}$$

$$= \|s(x) - s^*(x) + \Delta\|^2$$
$$= \|s(x) - s^*(x)\|^2 + 2\langle s(x) - s^*(x), \Delta \rangle + \|\Delta\|^2. \tag{8}$$

The third term does not depend on $s$, and so does not affect the minimizer of this loss function. Also, for every $x$, the second term is zero on average over $(z \mid x)$, so we have

$$\arg\min_s \mathbb{E}_{x,z}[l(s,x,z)] = \arg\min_s \mathbb{E}_{x,z}[\|s(x) - s^*(x)\|^2]$$

This shows that the score matching objective is indeed minimized by the true score. Moreover, an $\varepsilon$-approximate optimizer of $l(s)$ will be close in $L^2$, as needed by prior samplers.

**Understanding the ERM.** The algorithm chooses the score function $s$ minimizing the empirical loss,

$$\hat{\mathbb{E}}_{x,z}[l(s,x,z)] \coloneqq \frac{1}{m} \sum_{i=1}^m l(s, x_i, z_i)$$
$$= \hat{\mathbb{E}}_{x,z}[\|s(x) - s^*(x)\|^2 + 2\langle s(x) - s^*(x), \Delta \rangle + \|\Delta\|^2].$$

Again, the $\hat{\mathbb{E}}[\|\Delta\|^2]$ term is independent of $s$, so it has no effect on the minimizer and we can drop it from the loss function. We thus define

$$l'(s,x,z) \coloneqq \|s(x) - s^*(x)\|^2 + 2\langle s(x) - s^*(x), \Delta \rangle \tag{9}$$

that satisfies $l'(s^*, x, z) = 0$ and $\mathbb{E}[l'(s,x,z)] = \mathbb{E}[\|s(x) - s^*(x)\|^2]$. Our goal is now to to show that

$$\hat{\mathbb{E}}_{x,z}[l'(s,x,z)] > 0 \tag{10}$$

for all candidate score functions $s$ that are "far" from $s^*$. This would ensure that the empirical minimizer of the score matching objective is not "far" from $s^*$. To do this, we show that (10) is true with high probability for each individual $s$, then take a union bound over a net.

**Boundedness of $\Delta$.** Now, $z \sim N(0, \sigma^2 I_d)$ is technically unbounded, but is exponentially close to being bounded: $\|z\| \lesssim \sigma\sqrt{d}$ with overwhelming probability. So for the purpose of this proof overview, imagine that $z$ were drawn from a distribution of bounded norm, i.e., $\|z\| \le B\sigma$ always; the full proof (given in Appendix A) needs some exponentially small error terms to handle the tiny mass the Gaussian places outside this ball. Then, since $\Delta = \frac{z}{\sigma^2} - \mathbb{E}_{z|x}[\frac{z}{\sigma^2}]$, we get $\|\Delta\| \le 2B/\sigma$.

**Warmup:** $\mathrm{poly}(R/\sigma)$. As a warmup, consider the setting of prior work [BMR20]: (1) $\|x\| \le R$ always, so $\|s^*(x)\| \lesssim \frac{R}{\sigma^2}$; and (2) we only optimize over candidate score functions $s$ with value clipped to within $O(\frac{R}{\sigma^2})$, so $\|s(x) - s^*(x)\| \lesssim \frac{R}{\sigma^2}$. With both these restrictions, then, $|l'(s,x,z)| \le \frac{R^2}{\sigma^4} + \frac{RB}{\sigma^3}$. We can then apply a Chernoff bound to show concentration of $l'$: for $\mathrm{poly}(\varepsilon, \frac{R}{\sigma}, B, \log\frac{1}{\delta_{\mathrm{train}}})$ samples, with $1 - \delta_{\mathrm{train}}$ probability we have

$$\hat{\mathbb{E}}_{x,z}[l'(s,x,z)] \ge \mathbb{E}_{x,z}[l'(s,x,z)] - \frac{\varepsilon}{\sigma^2}$$
$$= \mathbb{E}[\|s(x) - s^*(x)\|^2] - \frac{\varepsilon^2}{\sigma^2}$$

which is greater than zero if $\mathbb{E}[\|s(x) - s^*(x)\|^2] > \frac{\varepsilon^2}{\sigma^2}$. Thus the ERM would reject each score function that is far in $L^2$. However, as we show in Section 4, restrictions (1) and (2) are both necessary: the score matching ERM needs a polynomial dependence on both the distribution norm and the candidate score function values to learn in $L^2$.

The main technical contribution of our paper is to avoid this polynomial dependence on $1/\sigma$. To do so, we settle for rejecting score functions $s$ that are far in our stronger distance measure $D_p^{\delta_{\mathrm{score}}}$, i.e., for which

$$\mathbb{P}[\|s(x) - s^*(x)\| > \varepsilon/\sigma] \ge \delta_{\mathrm{score}}. \tag{11}$$

**Approach.** We want to show (10), which is a concentration over $x$ and $z$. Now, $x$ is somewhat hard to control, because it depends on the unknown distribution, but $z \sim N(0, \sigma^2 I_d)$ is very well behaved. This motivates breaking up the expectation over $x, z$ into an expectation over $x$ and $z \mid x$. Following this approach, we could try to show that

$$\hat{\mathbb{E}}_{x,z}[l'(s, x, z)] \geq \hat{\mathbb{E}}_x[\mathbb{E}_{z|x}[l'(s, x, z)]]$$

However, this is not possible to show; the problem is that this could be unbounded, since $s(x)$ is an arbitrary neural network that could have extreme outliers. So, we instead show this is true with high probability if we clip the internal value, making it

$$A_x := \hat{\mathbb{E}}_x[\min(\mathbb{E}_{z|x}[l'(s, x, z)], \frac{10B^2}{\sigma^2})] = \hat{\mathbb{E}}_x[\min(\|s(x) - s^*(x)\|^2, \frac{10B^2}{\sigma^2})].$$

Note that $A_x$ is a function of the empirical samples $x$. If $s$ is a score that we want to reject under (11), then we know that $A_x$ is an empirical average of values that are at least $\frac{\varepsilon^2}{\sigma^2}$ with probability $\delta_{\text{score}}$. It therefore holds that, for $m > O(\frac{\log \frac{1}{\delta_{\text{train}}}}{\delta_{\text{score}}})$, we will with $1 - \delta_{\text{train}}$ probability over the samples $x$ have

$$A_x \gtrsim \frac{\varepsilon^2 \delta_{\text{score}}}{\sigma^2}. \tag{12}$$

**Concentration about the intermediate notion.** Finally, we show that for *every* set of samples $x_i$ satisfying (12), we will have

$$\hat{\mathbb{E}}_{z|x}[l'(s, x, z)] \geq \frac{A_x}{2} > 0.$$

This then implies

$$\hat{\mathbb{E}}_{x,z}[l'(s, x, z)] \geq \hat{\mathbb{E}}_x\left[\frac{A_x}{2}\right] \gtrsim \frac{\varepsilon^2 \delta_{score}}{\sigma^2},$$

as needed. For each sample $x$, we split our analysis of $\hat{E}_{z|x}[l(s, x, z)]$ into two cases:

**Case 1:** $\|s(x) - s^*(x)\| > O(\frac{B}{\sigma})$. In this case, by Cauchy-Schwarz and the assumption that $\|\Delta\| \leq 2B/\sigma$,

$$l'(s, x, z) \geq \|s(x) - s^*(x)\|^2 - O(\frac{B}{\sigma})\|s(x) - s^*(x)\| \geq \frac{10B^2}{\sigma^2}$$

so these $x$ will contribute the maximum possible value to $A_x$, regardless of $z$ (in its bounded range).

**Case 2:** $\|s(x) - s^*(x)\| < O(\frac{B}{\sigma})$. In this case, $|l'(s, x, z)| \lesssim B^2/\sigma^2$ and

$$\text{Var}_{z|x}(l'(s, x, z)) = 4\mathbb{E}[\langle s(x) - s^*(x), \Delta \rangle^2] \lesssim \frac{B^2}{\sigma^2}\|s(x) - s^*(x)\|^2$$

so for these $x$, as a distribution over $z$, $l'$ is bounded with bounded variance.

In either case, the contribution to $A_x$ is bounded with bounded variance; this lets us apply Bernstein's inequality to show, if $m > O(\frac{B^2 \log \frac{1}{\delta_{\text{train}}}}{\sigma^2 A}) \approx O(\frac{B^2 \log \frac{1}{\delta_{\text{train}}}}{\varepsilon^2 \delta_{score}})$, for every $x$ we will have

$$\hat{\mathbb{E}}_z[l'(s, x, z)] \geq \frac{A}{2} > 0$$

with $1 - \delta_{\text{train}}$ probability.

**Conclusion.** Suppose $m > O(\frac{B^2 \log \frac{1}{\delta_{\text{train}}}}{\varepsilon^2 \delta_{\text{score}}})$. Then with $1 - \delta_{\text{train}}$ probability we will have (12); and conditioned on this, with $1 - \delta_{\text{train}}$ probability we will have $\hat{\mathbb{E}}_{x,z}[l'(s, x, z)] > 0$. Hence this $m$ suffices to distinguish any candidate score $s$ that is far from $s^*$. For finite hypothesis classes we can take the union bound, incurring a $\log |\mathcal{H}|$ loss (this is given as Theorem A.2). Lemma 1.4 follows from applying this to a net over neural networks, which has size $\log H \approx PD \log \Theta$.

## 3.2 Sampling

Now we overview the proof of Lemma 1.3, i.e., why having an $\varepsilon/\sigma_{T-t_k}$ accuracy in the $D_q^\delta$ sense is sufficient for accurate sampling.

To practically implement the reverse SDE in (2), we discretize this process into $N$ steps and choose a sequence of times $0 = t_0 < t_1 < \cdots < t_N < T$. At each discretization time $t_k$, we use our score estimates $\widehat{s}_{t_k}$ and proceed with an *approximate* reverse SDE using our score estimates, given by the following. For $t \in [t_k, t_{k+1}]$,

$$\mathrm{d}x_{T-t} = (x_{T-t} + 2\widehat{s}_{T-t_k}(x_{T-t_k}))\,\mathrm{d}t + \sqrt{2}\,\mathrm{d}B_t, \quad x_T \sim \mathcal{N}(0, I_d). \tag{13}$$

This is *almost* the reverse SDE that would give exactly correct samples, with two sources of error: it starts at $\mathcal{N}(0, I_d)$ rather than $q_T$, and it uses $\widehat{s}_{T-t_k}$ rather than the $s_{T-t}$. The first error is negligible, since $q_T$ is $e^{-T}$-close to $\mathcal{N}(0, I_d)$. But how much error does the switch from $s$ to $\widehat{s}$ introduce?

Let $Q$ be the law of the reverse SDE using $s$, and let $\widehat{Q}$ be the law of (13). Then Girsanov's theorem states that the distance between $Q$ and $\widehat{Q}$ is defined by the $L^2$ error of the score approximations:

$$\mathsf{KL}(Q \parallel \widehat{Q}) = \sum_{k=0}^{N-1} \mathop{\mathbb{E}}_{Q}\left[\int_{T-t_{k+1}}^{T-t_k} \|s_{T-t}(x_{T-t}) - \widehat{s}_{T-t_k}(x_{T-t_k})\|^2\right]$$

$$\approx \sum_{k=0}^{N-1} \mathop{\mathbb{E}}_{x \sim q_{T-t_k}}\left[\|s_{T-t_k}(x) - \widehat{s}_{T-t_k}(x)\|^2\right](t_{k+1} - t_k) \tag{14}$$

The first line is an *equality*. The second line comes from approximating $x_{T-t}$ by $x_{T-t_k}$, which for small time steps is quite accurate. So in previous work, good $L^2$ approximations to the score mean (14) is small, and hence $\mathsf{KL}(Q \parallel \widehat{Q})$ is small. In our setting, where we *cannot* guarantee a good $L^2$ approximation, Girsanov actually implies that we *cannot* guarantee that $\mathsf{KL}(Q \parallel \widehat{Q})$ is small.

However, since we finally hope to show closeness in $\mathsf{TV}$, we can circumvent the above as follows. We define an event $E$ to be the event that the score is bounded well at all time steps $x_{T-t_k}$, i.e.,

$$E := \bigwedge_{k \in \{1, \ldots, N\}} \left(\|\widehat{s}_{T-t_k}(x_{T-t_k}) - s_{T-t_k}(x_{T-t_k})\| \leq \frac{\varepsilon}{\sigma_{T-t_k}}\right).$$

If we have a $D_{q_{t_k}}^{\delta/N}$ accuracy for each score, we have $1 - \mathbb{P}[E] \leq \delta$. Therefore, if we look at the $\mathsf{TV}$ error between $Q$ and $\widehat{Q}$, instead of bounding $\mathsf{KL}(Q \parallel \widehat{Q})$. The $\mathsf{TV}$ between $Q$ and $\widehat{Q}$ can be then bounded by

$$\mathsf{TV}(Q, \widehat{Q}) \leq (1 - \mathbb{P}[E]) + \mathsf{TV}((Q \mid E), (\widehat{Q} \mid E)).$$

The second term $\mathsf{TV}((Q \mid E), (\widehat{Q} \mid E))$ is bounded because after conditioning on $E$, the score error is always bounded, so now we can use (14) to bound the KL divergence between $Q$ and $\widehat{Q}$, then use Pinsker's inequality to translate KL into TV distance.

## 4 Hardness of Learning in $L^2$

In this section, we give concrete examples where it is difficult to learn the score in $L^2$, even though learning to sufficient accuracy for sampling is possible. Previous works, such as [BBDD24], require the $L^2$ error of the score estimate $s_t$ to be bounded by $\varepsilon/\sigma_t$. We demonstrate that achieving this guarantee is prohibitively expensive: sampling from a $\sigma_t$-smoothed distribution requires at least $\mathrm{poly}(1/\sigma_t)$ samples. Thus, sampling from a distribution $\gamma$-close in 2-Wasserstein to $q_0$ requires polynomially many samples in $\frac{1}{\gamma}$.

To show this, we demonstrate two lower bound instances. Both of these instances provide a pair of distributions that are hard to distinguish in $L^2$, and emphasize different aspects of this hardness:

1. The first instance shows that even with a polynomially bounded set of distributions, it is *information theoretically impossible* to learn a score with small $L^2$ error with high probability, with fewer than $\mathrm{poly}(1/\gamma)$ samples.

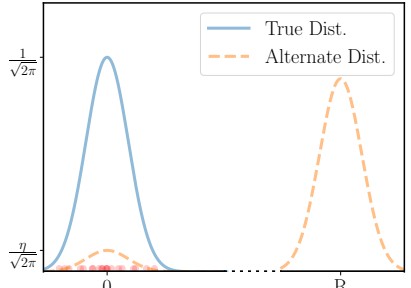 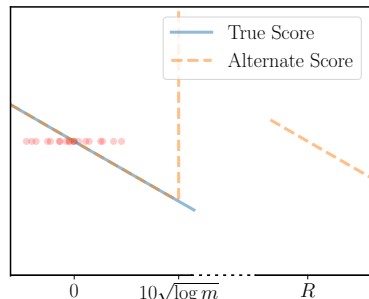

Figure 2: For $m$ samples from $\mathcal{N}(0,1)$, consider the score $\widehat{s}$ of the mixture $\eta\mathcal{N}(0,1) + (1-\eta)\mathcal{N}(R,1)$ above with $\eta$ is chosen so that $\widehat{s}(10\sqrt{\log m}) = 0$. For this $\widehat{s}$, the score-matching objective is close to 0, while the squared $L^2$ error is $\Omega\left(\frac{R^2}{m}\right)$.

2. In the second instance, we show that even with a simple true distribution, such as a single Gaussian, distinguishing the score matching loss of the true score function from one with high $L^2$ error can be challenging with fewer than $\text{poly}(1/\gamma)$ samples if the hypothesis class is large, such as with neural networks.

**Information theoretically indistinguishable distributions:** For our first example, consider the two distributions $(1-\eta)\mathcal{N}(0,\sigma^2) + \eta\mathcal{N}(\pm R, \sigma^2)$, where $R$ is polynomially large. Even though these distributions are polynomially bounded, it is *impossible* to distinguish these in $L^2$ given significantly fewer than $\frac{1}{\eta}$ samples. However, the $L^2$ error in score incurred from picking the score of the wrong distribution is large. In Figure 1, the rightmost plot shows a simulation of this example, and demonstrates that the $L^2$ error remains large even after many samples are taken. Formally, we have:

**Lemma 4.1.** *Let $R$ be sufficiently larger than $\sigma$. Let $p_1$ be the distribution $(1-\eta)\mathcal{N}(0,\sigma^2) + \eta\mathcal{N}(-R,\sigma^2)$ with corresponding score function $s_1$, and let $p_2$ be $(1-\eta)\mathcal{N}(0,\sigma^2) + \eta\mathcal{N}(R,\sigma^2)$ with score $s_2$, such that $\eta = \frac{\varepsilon^2\sigma^2}{R^2}$. Then, given $m < \frac{R^2}{\varepsilon^2\sigma^2}$ samples from either distribution, it is impossible to distinguish between $p_1$ and $p_2$ with probability larger than $1/2 + o_m(1)$. But,*

$$\mathbb{E}_{x\sim p_1}\left[\|s_1(x) - s_2(x)\|^2\right] \gtrsim \frac{\varepsilon^2}{\sigma^2} \qquad and \qquad \mathbb{E}_{x\sim p_2}\left[\|s_1(x) - s_2(x)\|^2\right] \gtrsim \frac{\varepsilon^2}{\sigma^2}.$$

**Simple true distribution:** Now, consider the true distribution being $N(0,\sigma^2)$, and, for large $S$, let $\widehat{s}$ be the score of the mixture distribution $\eta\mathcal{N}(0,\sigma^2) + (1-\eta)\mathcal{N}(S,\sigma^2)$, as in Figure 2. This score will have practically the same score matching objective as the true score for the given samples with high probability, as shown in Figure 2, since all $m$ samples will occur in the region where the two scores are nearly identical. However, the squared $L^2$ error incurred from picking the wrong score function $\widehat{s}$ is large We formally show this result in the following lemma:

**Lemma 4.2.** *Let $S$ be sufficiently large. Consider the distribution $\widehat{p} = \eta\mathcal{N}(0,\sigma^2) + (1-\eta)\mathcal{N}(S,\sigma^2)$ for $\eta = \frac{Se^{-\frac{S^2}{2} + 10\sqrt{\log m}\cdot S}}{10\sqrt{\log m}}$, and let $\widehat{s}$ be its score function. Given $m$ samples from the standard Gaussian $p^* = \mathcal{N}(0,\sigma^2)$ with score function $s^*$, with probability at least $1 - \frac{1}{poly(m)}$,*

$$\widehat{\mathbb{E}}\left[\|\widehat{s}(x) - s^*(x)\|^2\right] \leq \frac{1}{\sigma^2}e^{-O(S\sqrt{\log m})} \quad but \quad \mathbb{E}_{x\sim p^*}\left[\|\widehat{s}(x) - s^*(x)\|^2\right] \gtrsim \frac{S^2}{m\sigma^4}.$$

Together, these examples show that even with reasonably bounded or well-behaved distributions, it is difficult to learn the score in $L^2$ with fewer than $\text{poly}(R/\gamma)$ samples, motivating our $(1-\delta)$-quantile error measure.

# 5 Conclusion and Future Work

In this work, we have addressed the sample complexity of training the scores in diffusion models. We showed that a neural networks, when trained using the standard score matching objective, can

be used for DDPM sampling after $\widetilde{O}(\frac{d^2 PD}{\varepsilon^3} \log \Theta \log^3 \frac{1}{\gamma})$ training samples. This is an exponentially better dependence on the neural network depth $D$ and Wasserstein error $\gamma$ than given by prior work. To achieve this, we introduced a more robust measure, the $1 - \delta$ quantile error, which allows for efficient training with $\mathrm{poly}(\log \frac{1}{\gamma})$ samples using score matching. By using this measure, we showed that standard training (by score matching) and sampling (by the reverse SDE) algorithms achieve our new bound.

One caveat is that our results, as well as those of the prior work [BMR20] focus on understanding the *statistical* performance of the score matching objective: we show that the *empirical minimizer* of the score matching objective over the class of ReLU networks approximates the score accurately. We do not analyze the performance of Stochastic Gradient Descent (SGD), commonly used to *approximate* this empirical minimizer in practice. Understanding why SGD over the class of neural networks performs well is perhaps the biggest problem in theoretical machine learning, and we do not address it here.

## Acknowledgments

We thank Syamantak Kumar and Dheeraj Nagaraj for pointing out an error in a previous version of this work, and for providing a fix (Lemma F.7). This project was funded by NSF award CCF-1751040 (CAREER) and the NSF AI Institute for Foundations of Machine Learning (IFML).

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

# A   Sample Complexity to Achieve $(1 - \delta)$-Quantile Accuracy

The goal of this section is to prove our main lemma, the sample complexity bound to learn score at a single time. More specifically, we will give a quantitative version of Lemma 1.4.

**Lemma A.1** (Main Lemma, Quantitative Version). *Let $q_0$ be a distribution with second moment $m_2^2$. Let $\phi_\theta(\cdot)$ be the fully connected neural network with ReLU activations parameterized by $\theta$, with $P$ total parameters and depth $D$. Let $\Theta > 1$. Suppose there exists some weight vector $\theta^*$ with $\|\theta^*\|_\infty \leq \Theta$ such that*

$$\mathop{\mathbb{E}}_{x \sim q_t} \left[ \|\phi_{\theta^*}(x) - s_t(x)\|^2 \right] \leq \frac{\delta_{score} \cdot \delta_{train} \cdot \varepsilon^2}{C \cdot \sigma_t^2}$$

*for a sufficiently large constant $C$. By taking $m$ i.i.d. samples from $q_0$ to train score $s_t$, when*

$$m > \widetilde{O} \left( \frac{(d + \log \frac{1}{\delta_{train}}) \cdot PD}{\varepsilon^2 \delta_{score}} \cdot \log \left( \frac{\max(m_2, 1) \cdot \Theta}{\delta_{train}} \right) \right),$$

*the empirical minimizer $\phi_{\widehat{\theta}}$ of the score matching objective used to estimate $s_t$ (over $\phi_\theta$ with $\|\theta\|_\infty \leq \Theta$) satisfies*

$$D_{q_t}^{\delta_{score}}(\phi_{\widehat{\theta}}, s_t) \leq \varepsilon/\sigma_t.$$

*with probability $1 - \delta_{train}$.*

In order to prove the lemma, we first consider the case when learning the score over a finite function class $\mathcal{H}$.

## A.1   Score Estimation for Finite Function Class

The main result (Lemma A.2) of this section shows that if there is a function in the function class $\mathcal{H}$ that approximates the score well in our $D_p^\delta$ sense (see (6)), then the score-matching objective can learn this function using a number of samples that is independent of the domain size or the maximum value of the score.

**Notation.**   Fix a time $t$. For the purposes of this section, let $q := q_t$ be the distribution at time $t$, let $\sigma := \sigma_t$ be the smoothing level for time $t$, and let $s := s_t$ be the score function for time $t$. For $m$ samples $y_i \sim q_0$ and $z_i \sim \mathcal{N}(0, \sigma^2)$, let $x_i = e^{-t} y_i + z_i \sim q_t$.

We use $\widehat{\mathbb{E}}[f(x, y, z)]$ to denote the empirical expectation $\frac{1}{m} \sum_{i=1}^{m} f(x_i, y_i, z_i)$.

We now state the score matching algorithm.

---

**Algorithm 1** Empirical score estimation for $s$

---

**Input:** Distribution $q_0$, $y_1, \ldots, y_m \sim q_0$, set of hypothesis score function $\mathcal{H} = \{\tilde{s}_i\}$, smoothing level $\sigma$.

1. Take $m$ independent samples $z_i \sim N(0, \sigma^2 I_d)$, and let $x_i = e^{-t} y_i + z_i$.
2. For each $\tilde{s} \in \mathcal{H}$, let

$$l(\tilde{s}) = \frac{1}{m} \sum_{i=1}^{m} \left\| \tilde{s}(x_i) - \frac{-z_i}{\sigma^2} \right\|_2^2$$

3. Let $\hat{s} = \arg\min_{\tilde{s} \in \mathcal{H}} l(\tilde{s})$
4. Return $\hat{s}$

---

**Lemma A.2** (Score Estimation for Finite Function Class). *For any distribution $q_0$ and time $t > 0$, consider the $\sigma_t$-smoothed version $q_t$ with associated score $s_t$. For any finite set $\mathcal{H}$ of candidate score functions. If there exists some $s^* \in \mathcal{H}$ such that*

$$\mathbb{E}_{x \sim q_t} \left[ \|s^*(x) - s_t(x)\|_2^2 \right] \leq \frac{\delta_{score} \cdot \delta_{train} \cdot \varepsilon^2}{C \cdot \sigma_t^2}, \tag{15}$$

*for a sufficiently large constant $C$, then using $m > \widetilde{O} \left( \frac{1}{\varepsilon^2 \delta_{score}} (d + \log \frac{1}{\delta_{train}}) \log \frac{|\mathcal{H}|}{\delta_{train}} \right)$ samples, the empirical minimizer $\hat{s}$ of the score matching objective used to estimate $s_t$ satisfies*

$$D_{q_t}^{\delta_{score}}(\hat{s}, s_t) \leq \varepsilon/\sigma_t$$

*with probability $1 - \delta_{train}$.*

*Proof.* Per the notation discussion above, we set $s = s_t$ and $\sigma = \sigma_t$.

Denote

$$l(s, x, z) := \left\| s(x) - \frac{-z}{\sigma^2} \right\|_2^2$$

We will show that for all $\tilde{s}$ such that $D_q^{\delta_{score}}(\tilde{s}, s) > \varepsilon/\sigma$, with probability $1 - \delta_{train}$,

$$\hat{\mathbb{E}} \left[ l(\tilde{s}, x, z) - l(s^*, x, z) \right] > 0,$$

so that the empirical minimizer $\hat{s}$ is guaranteed to have

$$D_q^{\delta_{score}}(\hat{s}, s) \leq \varepsilon/\sigma.$$

We have

$$l(\tilde{s}, x, z) - l(s^*, x, z) = \left\| \tilde{s}(x) - \frac{-z}{\sigma^2} \right\|^2 - \left\| s^*(x) - \frac{-z}{\sigma^2} \right\|^2$$

$$= \left( \left\| \tilde{s}(x) - \frac{-z}{\sigma^2} \right\|^2 - \left\| s(x) - \frac{-z}{\sigma^2} \right\|^2 \right) - \|s^*(x) - s(x)\|^2 - 2\langle s^*(x) - s(x), s(x) - \frac{-z}{\sigma^2} \rangle. \tag{16}$$

Note that by Markov's inequality, with probability $1 - \delta_{train}/3$,

$$\widehat{\mathbb{E}}_x \left[ \|s^*(x) - s(x)\|^2 \right] \leq \frac{\delta_{score} \cdot \varepsilon^2}{30 \cdot \sigma^2}.$$

By Lemma A.3, with probability $1 - \delta_{train}/3$,

$$\widehat{\mathbb{E}} \left[ \langle s^*(x) - s(x), s(x) - \frac{-z}{\sigma^2} \rangle \right] \leq \frac{\delta_{score} \cdot \varepsilon^2}{100\sigma^2}$$

Also, by Corollary A.5, with probability $1 - \delta_{\text{train}}/3$, for all $\tilde{s} \in \mathcal{H}$ that satisfy $D_q^{\delta_{\text{score}}}(\tilde{s}, s) > \varepsilon/\sigma$ simultaneously,

$$\hat{\mathbb{E}}\left[\left\|\tilde{s}(x) - \frac{-z}{\sigma^2}\right\|^2 - \left\|s(x) - \frac{-z}{\sigma^2}\right\|^2\right] \geq \frac{\delta_{\text{score}}\varepsilon^2}{16\sigma^2}.$$

Plugging in everything into equation (16), we have, with probability $1 - \delta_{\text{train}}$, for all $\tilde{s} \in \mathcal{H}$ with $D_q^{\delta_{\text{score}}}(\tilde{s}, s) > \varepsilon/\sigma$ simultaneously,

$$\hat{\mathbb{E}}\left[l(\tilde{s}, x, z) - l(s^*, x, z)\right] \geq \frac{\delta_{\text{score}}\varepsilon^2}{16\sigma^2} - \frac{\delta_{\text{score}}\varepsilon^2}{30\sigma^2} - \frac{\delta_{\text{score}}\varepsilon^2}{50\sigma^2} > 0$$

as required. $\qquad\square$

**Lemma A.3.** *For any distribution $q_0$ and time $t > 0$, consider the $\sigma_t$-smoothed version $q_t$ with associated score $s_t$. Suppose $s^*$ is such that*

$$\mathop{\mathbb{E}}_{x \sim q_t}\left[\|s^*(x) - s(x)\|^2\right] \leq \frac{\delta_{score} \cdot \delta_{train} \cdot \varepsilon^2}{C\sigma_t^2}$$

*for a sufficiently large constant $C$. Then, using $m > \widetilde{O}\left(\frac{1}{\varepsilon^2 \delta_{score}}\left(d + \log \frac{1}{\delta_{train}}\right)\right)$ samples $(x_i, y_i, z_i)$ where $y_i \sim q_0$, $z_i \sim \mathcal{N}(0, \sigma^2 I_d)$ and $x_i = e^{-t}y_i + z_i \sim q_t$, we have with probability $1 - \delta_{train}$,*

$$\widehat{\mathbb{E}}\left[\langle s^*(x) - s(x), s(x) - \frac{-z}{\sigma^2}\rangle\right] \leq \frac{\delta_{score}\varepsilon^2}{1000\sigma^2}$$

*Proof.* Note that $s(x) = \mathbb{E}_{z|x}\left[\frac{-z}{\sigma^2}\right]$ so that

$$\mathbb{E}\left[\langle s^*(x) - s(x), s(x) - \frac{-z}{\sigma^2}\rangle\right] = 0$$

Also, for any $\delta$, with probability $1 - \delta$, by Lemmas F.4 and F.5,

$$\|s(x) - \frac{-z}{\sigma^2}\|^2 \leq C\frac{d + \log\frac{1}{\delta}}{\sigma^2}.$$

for some constant $C$. Let $E$ be the event that $\|s(x) - \frac{-z}{\sigma^2}\|^2 \leq C\frac{d + \log\frac{d}{\varepsilon^2 \delta_{\text{score}}\delta_{\text{train}}}}{\sigma^2}$. Since $\|s(x) - \frac{-z}{\sigma^2}\|^2$ is subexponential with parameter $\sigma^2$ and has mean $O\left(\frac{d}{\sigma^2}\right)$ by the above, by Lemma F.7, we have

$$\mathbb{E}\left[\|s(x) - \frac{-z}{\sigma^2}\|^2|\bar{E}\right] \lesssim \frac{d + \log\frac{d}{\varepsilon^2 \delta_{\text{score}}\delta_{\text{train}}}}{\sigma^2}$$

Then, since $\mathbb{P}[E] \geq 1 - \frac{\varepsilon^2 \delta_{\text{score}}\delta_{\text{train}}}{100(d + \log\frac{d}{\varepsilon^2 \delta_{\text{score}}\delta_{\text{train}}})} \geq \frac{1}{2}$ and $\mathbb{E}\left[\|s^*(x) - s(x)\|^2|\bar{E}\right] \leq \mathbb{E}\left[\|s^*(x) - s(x)\|^2\right]/\mathbb{P}[\bar{E}]$, we have

$$\mathbb{E}\left[\langle s^*(x) - s(x), s(x) - \frac{-z}{\sigma^2}\rangle|E\right]$$

$$= -\frac{\mathbb{E}\left[\langle s^*(x) - s(x), s(x) - \frac{-z}{\sigma^2}\rangle|\bar{E}\right]\mathbb{P}[\bar{E}]}{\mathbb{P}[E]}$$

$$\leq \frac{\sqrt{\mathbb{E}\left[\|s^*(x) - s(x)\|^2|\bar{E}\right]\mathbb{E}\left[\|s(x) - \frac{-z}{\sigma^2}\|^2|\bar{E}\right]}\mathbb{P}[\bar{E}]}{\mathbb{P}[E]}$$

$$\leq 2\sqrt{\mathbb{E}[\|s^*(x) - s(x)\|^2]\mathbb{E}\left[\|s(x) - \frac{-z}{\sigma^2}\|^2|\bar{E}\right]\mathbb{P}[\bar{E}]}$$

$$\lesssim \frac{1}{\sigma^2}\sqrt{\frac{\varepsilon^2 \delta_{\text{score}}\delta_{\text{train}}}{C}} \cdot \sqrt{d + \log\frac{d}{\varepsilon^2 \delta_{\text{score}}\delta_{\text{train}}}} \cdot \sqrt{\frac{\varepsilon^2 \delta_{\text{score}}\delta_{\text{train}}}{100(d + \log\frac{d}{\varepsilon^2 \delta_{\text{score}}\delta_{\text{train}}})}}$$

$$\leq \frac{\varepsilon^2 \delta_{\text{score}} \delta_{\text{train}}}{\sqrt{C}\sigma^2}.$$

Moreover,

$$\mathbb{E}\left[\langle s^*(x) - s(x), s(x) - \frac{-z}{\sigma^2}\rangle^2 | E\right] \leq \mathbb{E}\left[\|s^*(x) - s(x)\|^2 \|s(x) - \frac{-z}{\sigma^2}\|^2 | E\right]$$

$$\lesssim \mathbb{E}\left[\|s^*(x) - s(x)\|^2 (\frac{d + \log \frac{d}{\varepsilon^2 \delta_{\text{score}} \delta_{\text{train}}}}{\sigma^2}) | E\right]$$

$$\lesssim \frac{\delta_{\text{score}} \cdot \delta_{\text{train}} \cdot \varepsilon^2}{C\sigma^2} \cdot \frac{d + \log \frac{d}{\varepsilon^2 \delta_{\text{score}} \delta_{\text{train}}}}{\sigma^2}.$$

So, by Chebyshev's inequality, with probability $1 - \delta_{\text{train}}/2$,

$$\widehat{\mathbb{E}}\left[\langle s^*(x) - s(x), s(x) - \frac{-z}{\sigma^2}\rangle | E\right] \lesssim \frac{\varepsilon^2 \delta_{\text{score}} \delta_{\text{train}}}{\sqrt{C}\sigma^2} + \frac{1}{\sigma^2}\sqrt{\frac{\delta_{\text{score}} \cdot \varepsilon^2 \cdot (d + \log \frac{d}{\varepsilon^2 \delta_{\text{score}} \delta_{\text{train}}})}{Cm}} \lesssim \frac{\delta_{\text{score}} \cdot \varepsilon^2}{\sqrt{C}\sigma^2}$$

for our choice of $m$. Since $E$ holds except with probability $\frac{\varepsilon^2 \delta_{\text{score}} \delta_{\text{train}}}{100(d+\log \frac{d}{\varepsilon^2 \delta_{\text{score}} \delta_{\text{train}}})} \ll \delta_{\text{train}}$, we have with probability $1 - \delta_{\text{train}}$ in total,

$$\widehat{\mathbb{E}}\left[\langle s^*(x) - s(x), s(x) - \frac{-z}{\sigma^2}\rangle\right] \leq \frac{\delta_{\text{score}} \cdot \varepsilon^2}{1000\sigma^2}$$

for sufficiently large constant $C$. □

**Lemma A.4.** *Consider any set $\mathcal{F}$ of functions $f : \mathbb{R}^d \to \mathbb{R}^d$ such that for all $f \in \mathcal{F}$,*

$$\mathbb{P}_{x \sim p}\left[\|f(x)\| > \varepsilon/\sigma\right] > \delta_{score}.$$

*Then, with $m > \widetilde{O}\left(\frac{1}{\varepsilon^2 \delta_{score}}(d + \log \frac{1}{\delta_{train}})\log \frac{|\mathcal{F}|}{\delta_{train}}\right)$ samples drawn in Algorithm 1, we have with probability $1 - \delta_{train}$,*

$$\frac{1}{m}\sum_{i=1}^{m} -2\left(\frac{-z_i}{\sigma^2} - \mathbb{E}\left[\frac{-z}{\sigma^2}|x_i\right]\right)^T f(x_i) + \frac{1}{2}\|f(x_i)\|^2 \geq \frac{\delta_{score} \cdot \varepsilon^2}{16\sigma^2}$$

*holds for all $f \in \mathcal{F}$.*

*Proof.* Define

$$h_f(x, z) := -2\left(\frac{-z}{\sigma^2} - \mathbb{E}\left[\frac{-z}{\sigma^2}|x\right]\right)^T f(x) + \frac{1}{2}\|f(x)\|^2$$

We want to show that $h_f$ has

$$\widehat{\mathbb{E}}[h_f(x, z)] := \frac{1}{m}\sum_{i=1}^{m} h_f(x_i, z_i) \geq \frac{\delta_{\text{score}}\varepsilon^2}{16\sigma^2} \tag{17}$$

for all $f \in \mathcal{F}$ with probability $1 - \delta_{\text{train}}$.

Let $B = O\left(\frac{\sqrt{d + \log \frac{m}{\varepsilon \delta_{\text{score}} \delta_{\text{train}}}}}{\sigma}\right)$. For $f \in \mathcal{F}$, let

$$g_f(x, z) = \begin{cases} B^2 & \text{if } \|f(x)\| \geq 10B \\ h_f(x, z) & \text{otherwise} \end{cases}$$

be a clipped version of $h_f(x, z)$. We will show that for our chosen number of samples $m$, the following hold with probability $1 - \delta_{\text{train}}$ simultaneously:

1. For all $i$, $\left\|\frac{-z_i}{\sigma^2}\right\| \leq B$.

2. For all $i$, $\left\|\mathbb{E}[\frac{-z}{\sigma^2}|x_i]\right\| \leq B$

3. $\hat{\mathbb{E}}[g_f(x,z)] \geq \frac{\delta_{\text{score}}\varepsilon^2}{16}$ for all $f \in \mathcal{F}$

To show that these together imply (17), note that whenever $g_f(x_i, z_i) \neq h_f(x_i, z_i)$, $\|f(x_i)\| \geq 10B$. So, since $\|\frac{-z_i}{\sigma^2}\| \leq B$ and $\|\mathbb{E}[\frac{-z}{\sigma^2}|x_i]\| \leq B$,

$$h_f(x_i, z_i) = -2\left(\frac{-z_i}{\sigma^2} - \mathbb{E}\left[\frac{-z}{\sigma^2}|x_i\right]\right)^T f(x_i) + \frac{1}{2}\|f(x_i)\|^2 \geq -4B\|f(x_i)\| + \frac{1}{2}\|f(x_i)\|^2 \geq B^2 \geq g_f(x_i, z_i).$$

So under conditions $1, 2, 3$, for all $f \in \mathcal{F}$,

$$\hat{\mathbb{E}}[h_f(x,z)] \geq \hat{\mathbb{E}}[g_f(x,z)] \geq \frac{\delta_{\text{score}}\varepsilon^2}{16\sigma^2}$$

So it just remains to show that conditions $1, 2, 3$ hold with probability $1 - \delta_{\text{train}}$ simultaneously.

1. **For all i, $\left\|\frac{-z_i}{\sigma^2}\right\| \leq B$.** Holds with probability $1 - \delta_{\text{train}}/3$ by Lemma F.5 and the union bound.

2. **For all i, $\left\|\mathbb{E}\left[\frac{-z}{\sigma^2} \mid x_i\right]\right\| \leq B$.** Holds with probability $1 - \delta_{\text{train}}/3$ by Lemma F.6 and the union bound.

3. **$\hat{\mathbb{E}}[g_f(x,z)] \geq \frac{\delta_{\text{score}}\varepsilon^2}{16\sigma^2}$ for all $f \in \mathcal{F}$.**

   Let $E$ be the event that 1. and 2. hold. Let $a_i = \min(\|f(x_i)\|, 10B)$. We proceed in multiple steps.

   - Conditioned on $E$, $|g_f(x_i, z_i)| \lesssim B^2$.
     If $\|f(x_i)\| \geq 10B$, $|g_f(x_i, z_i)| = B^2$ by definition. On the other hand, when $\|f(x_i)\| < 10B$, since we condition on $E$,

     $$|g_f(x_i, z_i)| = |h_f(x_i, z_i)| = \left|-2\left(\frac{-z_i}{\sigma^2} - \mathbb{E}\left[\frac{-z}{\sigma^2}|x_i\right]\right)^T f(x_i) + \frac{1}{2}\|f(x_i)\|^2\right| \lesssim B^2$$

   - $\mathbb{E}[g_f(x_i, z_i)|E, a_i] \gtrsim a_i^2 - O(\delta_{\text{train}}B^2)$.
     First, note that by definition of $g_f(x, z)$, for $a_i = 10B$,

     $$\mathbb{E}[g_f(x_i, z_i)|a_i = 10B] = B^2$$

   Now, for $a_i < 10B$,

   $$\mathbb{E}[g_f(x_i, z_i)|a_i] = \mathbb{E}[h_f(x_i, z_i)|a_i]$$
   $$= \mathbb{E}_{x_i \mid \|f(x_i)\| = a_i}\left[\mathbb{E}_{-z|x_i}[h_f(x_i, z)]\right]$$

   Now, note that

   $$\mathbb{E}_{z|x}[h_f(x, z)] = \frac{1}{2}\|f(x)\|^2$$

   So, for $a < 10B$

   $$\mathbb{E}[g_f(x_i, z_i)|a_i] = \frac{1}{2}a_i^2$$

   Now let $g_f^{\text{clip}}(x_i, z_i)$ be a clipped version of $g_f(x_i, z_i)$, clipped to $\pm CB^2$ for sufficiently large constant $C$. We have, by above,

   $$\mathbb{E}[g_f(x_i, z_i)|a_i, E] = \mathbb{E}[g_f^{\text{clip}}(x_i, z_i)|a_i, E]$$

   But,

   $$\mathbb{E}[g_f^{\text{clip}}(x_i, z_i)|a_i, E] \geq \mathbb{E}[g_f^{\text{clip}}(x_i, z_i)|a_i] - O(\delta_{\text{train}}B^2)$$
   $$\gtrsim a_i^2 - O(\delta_{\text{train}}B^2)$$

- $\text{Var}(g_f(x_i, z_i)|a_i, E) \lesssim a_i^2 B^2$.

  For $a_i = 10B$, we have, by definition of $g_f(x, z)$,

  $$\text{Var}(g_f(x_i, z_i)|a_i, E) \lesssim B^4 \lesssim a_i^2 B^2$$

  On the other hand, for $a_i < 10B$,

  $$\text{Var}(g_f(x_i, z_i)|a_i, E) \leq \mathbb{E}\left[g_f(x_i, z_i)^2|a_i, E\right]$$
  $$= \mathbb{E}\left[\left(-2\left(\frac{-z_i}{\sigma^2} - \mathbb{E}\left[\frac{-z}{\sigma^2}|x_i\right]\right)^T f(x_i) + \frac{1}{2}\|f(x_i)\|^2\right)^2\right]$$
  $$\lesssim a_i^2 B^2$$

  by Cauchy-Schwarz.

- With probability $1 - \delta_{\text{train}}/3$, for all $f \in \mathcal{F}$, $\widehat{\mathbb{E}}[g_f(x_i, z_i)] \gtrsim \Omega\left(\frac{\varepsilon^2 \delta_{\text{score}}}{\sigma^2}\right)$

  Using the above, by Bernstein's inequality, with probability $1 - \delta_{\text{train}}/6$,

  $$\widehat{\mathbb{E}}\left[g_f(x_i, z_i)|a_i, E\right] \gtrsim \frac{1}{n}\sum_{i=1}^{n} a_i^2 - O(\delta_{\text{train}} B^2) - \frac{1}{n}B\sqrt{\sum_{i=1}^{n} a_i^2 \log\frac{1}{\delta_{\text{train}}}} - \frac{1}{n}B^2 \log\frac{1}{\delta_{\text{train}}}$$

  Now, note that since $\mathbb{P}_{x \sim p_\sigma}\left[\|f(x)\| > \varepsilon/\sigma\right] \geq \delta_{\text{score}}$, we have with probability $1 - \delta_{\text{train}}/6$, for $n > O\left(\frac{\log\frac{1}{\delta_{\text{train}}}}{\delta_{\text{score}}}\right)$

  $$\frac{1}{n}\sum_{i=1}^{n} a_i^2 \geq \Omega\left(\frac{\varepsilon^2 \delta_{\text{score}}}{\sigma^2}\right)$$

  So, for $n > O\left(\frac{B^2 \cdot \sigma^2 \log\frac{1}{\delta_{\text{train}}}}{\varepsilon^2 \delta_{\text{score}}}\right)$, we have, with probability $1 - \delta_{\text{train}}/3$,

  $$\widehat{\mathbb{E}}\left[g_f(x_i, z_i)|E\right] \gtrsim \Omega\left(\frac{\varepsilon^2 \delta_{\text{score}}}{\sigma^2}\right) - O(\delta_{\text{train}} B^2)$$

  Rescaling so that $\delta_{\text{train}} \leq O\left(\frac{\varepsilon^2 \delta_{\text{score}}}{\sigma^2 \cdot B^2}\right)$, for $n > O\left(\frac{B^2 \cdot \sigma^2 \log\frac{B^2 \cdot \sigma^2}{\varepsilon^2 \delta_{\text{score}} \delta_{\text{train}}}}{\varepsilon^2 \cdot \delta \text{score}}\right)$, we have, with probability $1 - \delta_{\text{train}}/3$,

  $$\widehat{\mathbb{E}}\left[g_f(x_i, z_i)|E\right] \gtrsim \Omega\left(\frac{\varepsilon^2 \delta_{\text{score}}}{\sigma^2}\right)$$

  Combining with 1. and 2. gives the claim for a single $f \in \mathcal{F}$. Union bounding over the size of $\mathcal{F}$ gives the claim.

$\square$

**Corollary A.5.** *Let $\mathcal{H}_{bad}$ be a set of score functions such that for all $\tilde{s} \in \mathcal{H}_{bad}$,*

$$D_q^{\delta_{score}}(\tilde{s}, s) > \varepsilon/\sigma.$$

*Then, for $m > \widetilde{O}\left(\frac{1}{\varepsilon^2 \delta_{score}}(d + \log\frac{1}{\delta_{train}})\log\frac{|\mathcal{H}_{bad}|}{\delta_{train}}\right)$ samples drawn by Algorithm 1, we have with probability $1 - \delta_{train}$,*

$$\widehat{\mathbb{E}}\left[\|\tilde{s}(x) - \frac{-z}{\sigma^2}\|^2 - \|s(x) - \frac{-z}{\sigma^2}\|^2\right] \geq \frac{\delta_{score}\varepsilon^2}{16\sigma^2}$$

*for all $\tilde{s} \in \mathcal{H}_{bad}$.*

*Proof.* We have, for $f(x) := \tilde{s}(x) - s(x)$,

$$\|\tilde{s}(x) - \frac{-z}{\sigma^2}\|^2 - \|s(x) - \frac{-z}{\sigma^2}\|^2 = \|f(x) + (s(x) - \frac{-z}{\sigma^2})\|^2 - \|s(x) - \frac{-z}{\sigma^2}\|^2$$

$$= \|f(x)\|^2 + 2(s(x) - \frac{-z}{\sigma^2})^T f(x)$$

$$= \|f(x)\|^2 - 2(\frac{-z}{\sigma^2} - \mathbb{E}[\frac{-z}{\sigma^2}|x])^T f(x)$$

since $s(x) = \mathbb{E}\left[\frac{-z}{\sigma^2}|x\right]$ by Lemma F.1. Then, by definition, for $s \in \mathcal{H}_{\text{bad}}$, for the associated $f$, $\mathbb{P}[\|f(x)\| > \varepsilon/\sigma] > \delta_{\text{score}}$. So, by Lemma A.4, the claim follows. $\square$

## A.2   Score Training for Neural Networks

Now we are ready to apply the finite function class to neural networks by a net argument. In particular, we first prove a version that uses a Frobenious norm to bound the weight vector.

**Lemma A.6.** *Let $q_0$ be a distribution with second moment $m_2^2$. Let $\phi_\theta(\cdot)$ be the fully connected neural network with ReLU activations parameterized by $\theta$, with $P$ total parameters and depth $D$. Let $\Theta > 1$. Suppose there exists some weight vector $\theta^*$ with $\|\theta^*\|_F \leq \Theta$ such that*

$$\mathbb{E}_{x \sim q_t} \left[\|\phi_{\theta^*}(x) - s_t(x)\|^2\right] \leq \frac{\delta_{score} \cdot \delta_{train} \cdot \varepsilon^2}{C \cdot \sigma_t^2}$$

*for a sufficiently large constant $C$. By taking $m$ i.i.d. samples from $q_0$ to train score $s_t$, when*

$$m > \widetilde{O}\left(\frac{(d + \log \frac{1}{\delta_{train}}) \cdot PD}{\varepsilon^2 \delta_{score}} \cdot \log\left(\frac{\max(m_2, 1) \cdot \Theta}{\delta_{train}}\right)\right),$$

*the empirical minimizer $\phi_{\widehat{\theta}}$ of the score matching objective used to estimate $s_t$ (over $\phi_\theta$ with $\|\theta\|_F \leq \Theta$) satisfies*

$$D_{q_t}^{\delta_{score}}(\phi_{\widehat{\theta}}, s_t) \leq \varepsilon/\sigma_t.$$

*with probability $1 - \delta_{train}$.*

*Proof.* Per the notation discussion above, we set $s = s_t$ and $\sigma = \sigma_t$. Note that $\sigma < 1$ since $\sigma_t^2 = 1 - e^{-2t}$.

For any function $f$ denote

$$l(f, x, z) := \left\|f(x) - \frac{-z}{\sigma^2}\right\|^2,$$

we will show that for every $\widetilde{\theta}$ with $\|\widetilde{\theta}\|_F \leq \Theta$ such that $D_q^{\delta_{score}}(\phi_{\widetilde{\theta}}, s) > \varepsilon/\sigma$, with probability $1 - \delta_{train}$,

$$\widehat{\mathbb{E}}\left[l(\phi_{\widetilde{\theta}}, x, z) - l(\phi_{\theta^*}, x, z)\right] > 0$$

so that the empirical minimizer $\phi_{\widehat{\theta}}$ is guaranteed to have

$$D_q^{\delta_{score}}(\phi_{\widehat{\theta}}, s) \leq \varepsilon/\sigma.$$

First, note that since the ReLU activation is contractive, the total Lipschitzness of $\phi_\theta$ is at most the product of the spectral norm of the weight matrices at each layer. For any $\theta$, consider $\widetilde{\theta}$ such that

$$\|\widetilde{\theta} - \theta\|_F \leq \frac{\tau}{\sigma D \Theta^{D-1}}$$

Let $M_1, \ldots, M_D$ be the weight matrices at each layer of the neural net $\phi_\theta$, and let $\widetilde{M}_1, \ldots, \widetilde{M}_D$ be the corresponding matrices of $\phi_{\widetilde{\theta}}$.

We now show that $\|\phi_{\widetilde{\theta}}(x) - \phi_\theta(x)\|$ is small, using a hybrid argument. Define $y_i$ to be the output of a neural network with weight matrices $M_1, \ldots, M_i, \widetilde{M}_{i+1}, \ldots, \widetilde{M}_D$ on input $x$, so $y_0 = \phi_{\widetilde{\theta}}(x)$ and $y_D = \phi_\theta(x)$. Then we have

$$\|y_i - y_{i+1}\| \le \|x\| \cdot \left( \prod_{j \le i} \|M_j\| \right) \cdot \left\| M_{i+1} - \widetilde{M}_{i+1} \right\| \cdot \left( \prod_{j > i+1} \left\| \widetilde{M}_j \right\| \right)$$

$$\le \|x\| \, \Theta^{D-1} \left\| \widetilde{\theta} - \theta \right\|_F$$

and so

$$\left\| \phi_{\widetilde{\theta}}(x) - \phi_\theta(x) \right\|_2 = \|y_0 - y_D\| \le \sum_{i=0}^{D-1} \|y_i - y_{i+1}\| \le \|x\| \, D\Theta^{D-1} \left\| \widetilde{\theta} - \theta \right\|_F \le \|x\| \cdot \tau/\sigma.$$

Note that the dimensionality of $\theta$ is $P$. So, we can construct a $\frac{\tau}{\sigma D \Theta^{D-1}}$-net $N$ over the set $\{\theta : \|\theta\|_F \le \Theta\}$ of size $O\left( \frac{\sigma D \Theta^{D-1}}{\tau} \right)^P$, so that for any $\theta$ with $\|\theta\|_F \le \Theta$, there exists $\widetilde{\theta} \in N$ with

$$\|\phi_{\widetilde{\theta}}(x) - \phi_\theta(x)\|_2 \le (\tau/\sigma) \cdot \|x\|$$

Let $\mathcal{H} = \{\phi_{\widetilde{\theta}} : \widetilde{\theta} \in N\}$. Then, we have that for every $\theta$ with $\|\theta\|_F \le \Theta$, there exists $h \in \mathcal{H}$ such that

$$\widehat{\mathbb{E}}\left[ \|h(x) - \phi_\theta(x)\|^2 \right] \le (\tau/\sigma)^2 \cdot \frac{1}{m} \sum_{i=1}^m \|x_i\|^2 \tag{18}$$

Now, choose any $\widetilde{\theta}$ with $\|\widetilde{\theta}\|_F \le \Theta$ and $D_q^{\delta_{\text{score}}}(\phi_{\widetilde{\theta}}, s) > \varepsilon/\sigma$, and let $\widetilde{h} \in \mathcal{H}$ satisfy the above for $\widetilde{\theta}$. We will set

$$\tau = \frac{C' \varepsilon^2 \delta_{\text{score}}}{\Theta^D \left( m \cdot \max(m_2^2, 1) + (d + \log \frac{m}{\delta_{\text{train}}}) \right)} \tag{19}$$

for small enough constant $C'$. So, $|\mathcal{H}| = O\left( \frac{\sigma D \Theta^{D-1}}{\tau} \right)^P < O\left( \frac{D\Theta^{2D-1}(m \cdot m_2^2 + (d + \log \frac{m}{\delta_{\text{train}}}))}{\varepsilon^2 \delta_{\text{score}}} \right)^P$, since $\sigma < 1$.

So, our final choice of $m$ satisfies $m > \widetilde{O}\left( \frac{1}{\varepsilon^2 \delta_{\text{score}}} \left( d + \log \frac{1}{\delta_{\text{train}}} \right) \log \frac{|\mathcal{H}|}{\delta_{\text{train}}} \right)$.

We have

$$l(\phi_{\widetilde{\theta}}, x, z) - l(\phi_{\theta^*}, x, z)$$

$$= \|\phi_{\widetilde{\theta}}(x) - \frac{-z}{\sigma^2}\|^2 - \|\phi_{\theta^*}(x) - \frac{-z}{\sigma^2}\|^2$$

$$= \|\phi_{\widetilde{\theta}}(x) - \widetilde{h}(x)\|^2 + 2\langle \phi_{\widetilde{\theta}}(x) - \widetilde{h}(x), \widetilde{h}(x) - \frac{-z}{\sigma^2}\rangle + \|\widetilde{h}(x) - \frac{-z}{\sigma^2}\|^2 \tag{20}$$

$$- \|s(x) - \frac{-z}{\sigma^2}\|^2 - \|\phi_{\theta^*}(x) - s(x)\|^2 - 2\langle \phi_{\theta^*}(x) - s(x), s(x) - \frac{-z}{\sigma^2}\rangle$$

Now, by Corollary A.5, for our choice of $m$, with probability $1 - \delta_{\text{train}}/4$ for every $h \in \mathcal{H}$ with $D_q^{\delta_{\text{score}}/2}(h, s) > \varepsilon/(2\sigma)$ simultaneously,

$$\widehat{\mathbb{E}}\left[ \|h(x) - \frac{-z}{\sigma^2}\|^2 - \|s(x) - \frac{-z}{\sigma^2}\|^2 \right] \ge \frac{\delta_{\text{score}} \varepsilon^2}{128\sigma^2}$$

By Markov's inequality, with probability $1 - \delta_{\text{train}}/4$,

$$\widehat{\mathbb{E}}\left[ \|\phi_{\theta^*}(x) - s(x)\|^2 \right] \le \frac{\delta_{\text{score}} \cdot \varepsilon^2}{250\sigma^2}$$

By Lemma A.3, with probability $1 - \delta_{\text{train}}/4$,

$$\widehat{\mathbb{E}}\left[ \langle \phi_{\theta^*}(x) - s(x), s(x) - \frac{-z}{\sigma^2}\rangle \right] \le \frac{\delta_{\text{score}} \varepsilon^2}{1000\sigma^2}$$

Plugging into (20) we have that with probability $1 - 3\delta_{\text{train}}/4$, for every $\widetilde{h}$ satisfying $D_q^{\delta_{\text{score}}/2}(\widetilde{h}, s) > \varepsilon/(2\sigma)$,

$$\widehat{\mathbb{E}}\left[l(\phi_{\widetilde{\theta}}, x, z) - l(\phi_{\theta^*}, x, z)\right] \geq \frac{\delta_{\text{score}}\varepsilon^2}{128\sigma^2} - \frac{\delta_{\text{score}} \cdot \varepsilon^2}{250\sigma^2} - 2 \cdot \frac{\delta_{\text{score}}\varepsilon^2}{1000\sigma^2} + 2\langle\phi_{\widetilde{\theta}}(x) - \widetilde{h}(x), \widetilde{h}(x) - \frac{-z}{\sigma^2}\rangle$$

$$\geq \frac{\delta_{\text{score}} \cdot \varepsilon^2}{600 \cdot \sigma^2} + 2\langle\phi_{\widetilde{\theta}}(x) - \widetilde{h}(x), \widetilde{h}(x) - \frac{-z}{\sigma^2}\rangle \tag{21}$$

Now, we will show that $D_q^{\delta_{\text{score}}/2}(\widetilde{h}, s) > \varepsilon/(2\sigma)$, as well as bound the last term above, for every $\widetilde{\theta}$ satisfying $D_q^{\delta_{\text{score}}}(\phi_{\widetilde{\theta}}, s) > \varepsilon/\sigma$ and $\widetilde{h} \in \mathcal{H}$ the rounding of $\phi_{\widetilde{\theta}}$.

By the fact that $q_0$ has second moment $m_2^2$, we have that with probability $1 - \delta$ over $x$,

$$\|x\| \leq \frac{m_2}{\sqrt{\delta}} + \sigma\left(\sqrt{d} + \sqrt{2\log\frac{1}{\delta}}\right)$$

Now since $D_q^{\delta_{\text{score}}}(\phi_{\widetilde{\theta}}, s) > \varepsilon/\sigma$, and $\|\widetilde{h}(x) - \phi_{\widetilde{\theta}}(x)\| \leq (\tau/\sigma) \cdot \|x\|$, we have, with probability at least $\delta_{\text{score}}/2$,

$$\|\widetilde{h}(x) - s(x)\| \geq \|\phi_{\widetilde{\theta}}(x) - s(x)\| - \|\widetilde{h}(x) - \phi_{\widetilde{\theta}}(x)\|$$

$$\geq \varepsilon/\sigma - (\tau/\sigma) \cdot \left(\frac{m_2}{\sqrt{\delta_{\text{score}}/2}} + \sigma\left(\sqrt{d} + \sqrt{2\log\frac{2}{\delta_{\text{score}}}}\right)\right)$$

$$\geq \varepsilon/(2\sigma)$$

for our choice of $\tau$ as in (19). So, we have shown that $D_q^{\delta_{\text{score}}/2}(\widetilde{h}, s) > \varepsilon/(2\sigma)$.

Finally, we bound the last term in (21) above. We have by (18) and a union bound, with probability $1 - \delta_{\text{train}}/8$,

$$\widehat{\mathbb{E}}\left[\|\widetilde{h}(x) - \phi_{\widetilde{\theta}}(x)\|^2\right] \lesssim (\tau/\sigma)^2 \cdot \left(\frac{m \cdot m_2^2}{\delta_{\text{train}}} + \sigma^2\left(d + \log\frac{m}{\delta_{\text{train}}}\right)\right)$$

$$\lesssim (\tau/\sigma)^2 \cdot \left(\frac{m \cdot m_2^2}{\delta_{\text{train}}} + \left(d + \log\frac{m}{\delta_{\text{train}}}\right)\right) \quad \text{since } \sigma < 1$$

$$\lesssim \frac{C'^2\varepsilon^4\delta_{\text{score}}^2}{\sigma^2\Theta^{2D}} \cdot \frac{1}{\frac{m \cdot \max(m_2^2, 1)}{\delta_{\text{train}}} + \left(d + \log\frac{m}{\delta_{\text{train}}}\right)} \quad \text{by (19)}$$

Similarly, with probability $1 - \delta_{\text{train}}/8$,

$$\widehat{\mathbb{E}}\left[\|\widetilde{h}(x) - \frac{-z}{\sigma^2}\|^2\right] \lesssim \Theta^D \cdot \left(\frac{m \cdot m_2^2}{\delta_{\text{train}}} + \sigma^2\left(d + \log\frac{m}{\delta_{\text{train}}}\right)\right) + \frac{1}{\sigma^2}\left(d + \log\frac{m}{\delta_{\text{train}}}\right)$$

So, putting the above together, with probability $1 - \delta_{\text{train}}/4$,

$$\widehat{\mathbb{E}}\left[\|\widetilde{h}(x) - \phi_{\widetilde{\theta}}(x)\|^2\right] \cdot \widehat{\mathbb{E}}\left[\|\widetilde{h}(x) - \frac{-z}{\sigma^2}\|^2\right] \lesssim \frac{C'^2\varepsilon^4\delta_{\text{score}}^2}{\sigma^2}\left(1 + \frac{1}{\sigma^2}\right)$$

$$\lesssim \frac{C'^2\varepsilon^4\delta_{\text{score}}^2}{\sigma^4} \quad \text{since } \sigma < 1$$

So, with probability $1 - \delta_{\text{train}}/4$, for some small enough constant $C'$,

$$\widehat{\mathbb{E}}\left[\langle\phi_{\widetilde{\theta}}(x) - \widetilde{h}(x), \widetilde{h}(x) - \frac{-z}{\sigma^2}\rangle\right] \geq -\sqrt{\widehat{\mathbb{E}}\left[\|\widetilde{h}(x) - \phi_{\widetilde{\theta}}(x)\|^2\right] \cdot \widehat{\mathbb{E}}\left[\|\widetilde{h}(x) - \frac{-z}{\sigma^2}\|^2\right]}$$

$$\geq -\frac{\delta_{\text{score}}\varepsilon^2}{2000 \cdot \sigma^2}$$

So finally, combining with (21), we have with probability $1 - \delta_{\text{train}}$

$$\widehat{\mathbb{E}} \left[ l(\phi_{\widetilde{\theta}}, x, z) - l(\phi_{\theta^*}, x, z) \right] \geq \frac{\delta_{\text{score}} \cdot \varepsilon^2}{1000 \cdot \sigma^2} > 0$$

So, we have shown that with probability $1 - \delta_{\text{train}}$, for every $\widetilde{\theta}$ with $\|\widetilde{\theta}\|_F \leq \Theta$ and $D_q^{\delta_{\text{score}}}(\phi_{\widetilde{\theta}}, s) > \varepsilon/\sigma$,

$$\widehat{E} \left[ l(\phi_{\widetilde{\theta}}, x, z) - l(\phi_{\theta^*}, x, z) \right] > 0$$

so that the empirical minimizer $\phi_{\widehat{\theta}}$ is guaranteed to have

$$D_q^{\delta_{\text{score}}}(\phi_{\widehat{\theta}}, s) \leq \varepsilon/\sigma.$$

$\square$

Then we have our main lemma as a direct corollary:

*Proof of Lemma A.1.* For every $\widetilde{\theta}$ with $\|\widetilde{\theta}\|_\infty \leq \Theta$, we have

$$\|\widetilde{\theta}\|_F \leq \sqrt{P} \|\widetilde{\theta}\|_\infty \leq \sqrt{P} \Theta.$$

Then, by applying Lemma A.6 directly, we prove the lemma. $\square$

# B  Sampling with our score estimation guarantee

In this section, we show that diffusion models can converge to the true distribution without necessarily adhering to an $L^2$ bound on the score estimation error. A high probability accuracy of the score is sufficient.

In order to simulate the reverse process of (1) in an actual algorithm, the time was discretized into $N$ steps. The $k$-th step ends at time $t_k$, satisfying $0 \leq t_0 < t_1 < \cdots < t_N = T - \gamma$. The algorithm stops at $t_N$ and outputs the final state $x_{T-t_N}$.

To analyze the reverse process run under different levels of idealness, we consider these four specific path measures over the path space $\mathcal{C}([0, T - \gamma]; \mathbb{R}^d)$:

- Let $Q$ be the measure for the process that

$$\mathrm{d}x_{T-t} = (x_{T-t} + 2s_{T-t}(x_{T-t})) \, \mathrm{d}t + \sqrt{2} \, \mathrm{d}B_t, \quad x_T \sim q_T.$$

- Let $Q_{\text{dis}}$ be the measure for the process that for $t \in [t_k, t_{k+1}]$,

$$\mathrm{d}x_{T-t} = (x_{T-t} + 2s_{T-t_k}(x_{T-t_k})) \, \mathrm{d}t + \sqrt{2} \, \mathrm{d}B_t, \quad x_T \sim q_T.$$

- Let $\overline{Q}$ be the measure for the process that for $t \in [t_k, t_{k+1}]$,

$$\mathrm{d}x_{T-t} = (x_{T-t} + 2\widehat{s}_{T-t_k}(x_{T-t_k})) \, \mathrm{d}t + \sqrt{2} \, \mathrm{d}B_t, \quad x_T \sim q_T.$$

- Let $\widehat{Q}$ be the measure for the process that for $t \in [t_k, t_{k+1}]$,

$$\mathrm{d}x_{T-t} = (x_{T-t} + 2\widehat{s}_{T-t_k}(x_{T-t_k})) \, \mathrm{d}t + \sqrt{2} \, \mathrm{d}B_t, \quad x_T \sim \mathcal{N}(0, I_d).$$

To summarize, $Q$ represents the perfect reverse process of (1), $Q_{\text{dis}}$ is the discretized version of $Q$, $\overline{Q}$ runs $Q_{\text{dis}}$ with an estimated score, and $\widehat{Q}$ starts $\overline{Q}$ at $\mathcal{N}(0, I_d)$ — effectively the actual implementable reverse process.

Recent works have shown that under the assumption that the estimated score function is close to the real score function in $L^2$, then the output of $\widehat{Q}$ will approximate the true distribution closely. Our next theorem shows that this assumption is in fact not required, and it shows that our score assumption can be easily integrated in a black-box way to achieve similar results.

**Lemma B.1** (Score Estimation guarantee). *Consider an arbitrary sequence of discretization times* $0 = t_0 < t_1 < \cdots < t_N = T - \gamma$, *and let* $\sigma_t := \sqrt{1 - e^{-2t}}$. *Assume that for each* $k \in \{0, \ldots, N-1\}$, *the following holds:*

$$D_{q_{T-t_k}}^{\delta/N}(\widehat{s}_{T-t_k}, s_{T-t_k}) \leq \frac{\varepsilon}{\sigma_{T-t_k}}.$$

*Then, we have*

$$\mathbb{P}_Q \left[ \sum_{k=0}^{N-1} \|\widehat{s}_{T-t_k}(x_{T-t_k}) - s_{T-t_k}(x_{T-t_k})\|_2^2 (t_{k+1} - t_k) \leq \varepsilon^2 \left( T + \log \frac{1}{\gamma} \right) \right] \geq 1 - \delta.$$

*Proof.* Since random variable $x_{T-t_k}$ follows distribution $q_{T-t_k}$ under $Q$, for each $k \in \{0, \ldots, N-1\}$, we have

$$\mathbb{P}_Q \left[ \|\widehat{s}_{T-t_k}(x_{T-t_k}) - s_{T-t_k}(x_{T-t_k})\| \leq \frac{\varepsilon}{\sqrt{1 - e^{-2(T-t_k)}}} \right] \geq 1 - \frac{\delta}{N}.$$

Using a union bound over all $N$ different $\sigma$ values, it follows that with probability at least $1 - \delta$ over $Q$, the inequality

$$\|\widehat{s}_{T-t_k}(x_{T-t_k}) - s_{T-t_k}(x_{T-t_k})\|_2^2 \leq \frac{\varepsilon^2}{1 - e^{-2(T-t_k)}}.$$

is satisfied for every $k \in \{0, \ldots, N-1\}$. Under this condition, we have

$$\sum_{k=0}^{N-1} \|\widehat{s}_{T-t_k}(x_{T-t_k}) - s_{T-t_k}(x_{T-t_k})\|_2^2 (t_{k+1} - t_k)$$

$$\leq \sum_{k=0}^{N-1} \frac{\varepsilon^2}{1 - e^{-2(T-t_k)}} (t_{k+1} - t_k)$$

$$\leq \sum_{k=0}^{N-1} \int_{t_k}^{t_{k+1}} \frac{\varepsilon^2}{1 - e^{-2(T-t_k)}} \, \mathrm{d}t$$

$$\leq \int_0^{T-\gamma} \frac{\varepsilon^2}{1 - e^{-2(T-t_k)}} \, \mathrm{d}t$$

$$\leq \varepsilon^2 \left( T + \log \frac{1}{\gamma} \right).$$

Hence, we find that

$$\mathbb{P}_Q \left[ \sum_{k=0}^{N-1} \|\widehat{s}_{T-t_k}(x_{T-t_k}) - s_{T-t_k}(x_{T-t_k})\|_2^2 (t_{k+1} - t_k) \leq \varepsilon^2 \left( T + \log \frac{1}{\gamma} \right) \right] \geq 1 - \delta.$$

$\square$

**Lemma B.2** (Score estimation error to TV). *Let $q$ be an arbitrary distribution. If the score estimation satisfies that*

$$\mathbb{P}_Q \left[ \sum_{k=0}^{N-1} \|\widehat{s}_{T-t_k}(x_{T-t_k}) - s_{T-t_k}(x_{T-t_k})\|_2^2 (t_{k+1} - t_k) \leq \varepsilon^2 \right] \geq 1 - \delta, \tag{22}$$

*then the output distribution $p_{T-t_N}$ of $\widehat{Q}$ satisfies*

$$\mathsf{TV}(q_\gamma, p_{T-t_N}) \lesssim \delta + \varepsilon + \mathsf{TV}(Q, Q_{dis}) + \mathsf{TV}(q_T, \mathcal{N}(0, I_d)).$$

*Proof.* We will start by bounding the $\mathsf{TV}$ distance between $Q_{\text{dis}}$ and $\overline{Q}$. We will proceed by defining $\widetilde{Q}$ and arguing that both $\mathsf{TV}(Q_{\text{dis}}, \widetilde{Q})$ and $\mathsf{TV}(\widetilde{Q}, \overline{Q})$ are small. By the triangle inequality, this will imply that $Q$ and $\overline{Q}$ are close in $\mathsf{TV}$ distance.

**Defining $\widetilde{Q}$.** For $k \in \{0, \ldots, N-1\}$, consider event

$$E_k := \left( \sum_{i=0}^{k} \|\widehat{s}_{T-t_i}(x_{T-t_i}) - s_{T-t_i}(x_{T-t_i})\|_2^2 (t_{i+1} - t_i) \le \varepsilon^2 \right),$$

which represents that the accumulated score estimation error along the path is at most $\varepsilon^2$ for a discretized diffusion process.

Given $E_k$, we define a version of $Q_{\text{dis}}$ that is forced to have a bounded score estimation error. Let $\widetilde{Q}$ over $\mathcal{C}((0,T], \mathbb{R}^d)$ be the law of a modified reverse process initialized at $x_T \sim q_T$, and for each $t \in [t_k, t_{k+1})$,

$$\mathrm{d}x_{T-t} = -(x_{T-t} + 2\widetilde{s}_{T-t_k}(x_{T-t_k})) \, \mathrm{d}t + \sqrt{2} \, \mathrm{d}B_t, \tag{23}$$

where

$$\widetilde{s}_{T-t_k}(x_{T-t_k}) := \begin{cases} s_{T-t_k}(x_{T-t_k}) & E_k \text{ holds,} \\ \widehat{s}_{T-t_k}(x_{T-t_k}) & E_k \text{ doesn't hold.} \end{cases}$$

This SDE guarantees that once the accumulated score error exceeds $\varepsilon_{score}^2$ ($E_k$ fails to hold), we switch from the true score to the estimated score. Therefore, we have that the following inequality always holds:

$$\sum_{k=0}^{N-1} \|\widetilde{s}_{T-t_k}(x_{T-t_k}) - \widehat{s}_{T-t_k}(x_{T-t_k})\|_2^2 (t_{k+1} - t_k) \le \varepsilon^2. \tag{24}$$

**$Q_{\text{dis}}$ and $\widetilde{Q}$ are close.** By (22), we have

$$\mathbb{P}_{Q_{\text{dis}}}[E_0 \wedge \cdots \wedge E_{N-1}] = \mathbb{P}_{Q_{\text{dis}}}[E_{N-1}] \ge \mathbb{P}_Q[E_{N-1}] - \mathsf{TV}(Q, Q_{\text{dis}}) \ge 1 - \delta - \mathsf{TV}(Q, Q_{\text{dis}}),$$

Note that when a path $(x_{T-t})_{t \in [0,t_N]}$ satisfies $E_0 \wedge \cdots \wedge E_{N-1}$, its probability under $\widetilde{Q}$ is at least its probability under $Q_{\text{dis}}$. Therefore, we have

$$\mathsf{TV}(Q_{\text{dis}}, \widetilde{Q}) \lesssim \delta + \mathsf{TV}(Q, Q_{\text{dis}}).$$

**$\widetilde{Q}$ and $\overline{Q}$ are close.** Inspired by [CCL$^+$23b], we utilize Girsanov's theorem (see Theorem F.8) to help bound this distance. Define

$$b_r := \sqrt{2}(\widetilde{s}_{T-t_k}(x_{T-t_k}) - \widehat{s}_{T-t_k}(x_{T-t_k})),$$

where $k$ is index such that $r \in [t_k, t_{k+1})$. We apply the Girsanov's theorem to $(\widetilde{Q}, (b_r))$. By Eq. (24), we have

$$\int_0^{t_N} \|b_r\|_2^2 \, \mathrm{d}r \le \sum_{k=0}^{N-1} \|\sqrt{2}(\widetilde{s}_{T-t_k}(x_{T-t_k}) - \widehat{s}_{T-t_k}(x_{T-t_k}))\|_2^2 (t_{k+1} - t_k) \le 2\varepsilon^2 < \infty.$$

This satisfies Novikov's condition and tells us that for

$$\mathcal{E}(\mathcal{L})_t = \exp\left( \int_0^t b_r \, \mathrm{d}B_r - \frac{1}{2} \int_0^t \|b_r\|_2^2 \, \mathrm{d}r \right),$$

under measure $\widetilde{Q}' := \mathcal{E}(\mathcal{L})_{t_N} \widetilde{Q}$, there exists a Brownian motion $(\widetilde{B}_t)_{t \in [0,t_N]}$ such that

$$\widetilde{B}_t = B_t - \int_0^t b_r \, \mathrm{d}r,$$

and thus for $t \in [t_k, t_{k+1})$,

$$\mathrm{d}\widetilde{B}_t = \mathrm{d}B_t + \sqrt{2}(\widetilde{s}_{T-t_k}(x_{T-t_k}) - \widehat{s}_{T-t_k}(x_{T-t_k})) \, \mathrm{d}t.$$

Plug this into (23) and we have that for $t \in [t_k, t_{k+1})$

$$\mathrm{d}x_{T-t} = -(x_{T-t} + 2\widehat{s}_{T-t_k}(x_{T-t_k})) \, \mathrm{d}t + \sqrt{2} \, \mathrm{d}\widetilde{B}_t, \quad x_T \sim q_T.$$

This equation depicts the distribution of $x$, and this exactly matches the definition of $\overline{Q}$. Therefore, $\overline{Q} = \widetilde{Q}' = \mathcal{E}(\mathcal{L})_{t_N} \widetilde{Q}$, and we have

$$\mathsf{KL}\left(\widetilde{Q}\middle\|\overline{Q}\right) = \underset{\widetilde{Q}}{\mathbb{E}}\left[\ln \frac{\mathrm{d}\widetilde{Q}}{\mathrm{d}\overline{Q}}\right] = \underset{\widetilde{Q}}{\mathbb{E}}\left[\ln \mathcal{E}(\mathcal{L})_{t_N}\right].$$

Then by using (24), we have

$$\underset{\widetilde{Q}}{\mathbb{E}}\left[\ln \mathcal{E}(\mathcal{L})_{t_N}\right] \lesssim \underset{\widetilde{Q}}{\mathbb{E}}\left[\sum_{k=0}^{N-1} \|\widetilde{s}_{T-t_k}(x_{T-t_k}) - \widehat{s}_{T-t_k}(x_{T-t_k})\|_2^2 (t_{k+1} - t_k)\right] \lesssim \varepsilon^2.$$

Therefore, we can apply Pinsker's inequality and get

$$\mathsf{TV}(\widetilde{Q}, \overline{Q}) \leq \sqrt{\mathsf{KL}\left(\widetilde{Q}\middle\|\overline{Q}\right)} \lesssim \varepsilon.$$

**Putting things together.**    Using the data processing inequality, we have

$$\mathsf{TV}(\overline{Q}, \widehat{Q}) \leq \mathsf{TV}(q_T, \mathcal{N}(0, I_d)).$$

Combining these results, we have

$$\mathsf{TV}(Q, \widehat{Q}) \leq \mathsf{TV}(Q, Q_{\mathrm{dis}}) + \mathsf{TV}(Q_{\mathrm{dis}}, \widetilde{Q}) + \mathsf{TV}(\widetilde{Q}, \overline{Q}) + \mathsf{TV}(\overline{Q}, \widehat{Q})$$
$$\lesssim \delta + \varepsilon + \mathsf{TV}(Q, Q_{\mathrm{dis}}) + \mathsf{TV}(q_T, \mathcal{N}(0, I_d)).$$

Since $q_\gamma$ is the distribution for $x_{T-t_N}$ under $Q$ and $p_{T-t_N}$ is the distribution for $x_{T-t_N}$ under $\widehat{Q}$, we have

$$\mathsf{TV}(q_\gamma, p_{T-t_N}) \leq \mathsf{TV}(Q, \widehat{Q}) \lesssim \delta + \varepsilon + \mathsf{TV}(Q, Q_{\mathrm{dis}}) + \mathsf{TV}(q_T, \mathcal{N}(0, I_d)).$$

$\square$

**Lemma B.3.** *Consider an arbitrary sequence of discretization times* $0 = t_0 < t_1 < \cdots < t_N = T - \gamma$. *Assume that for each* $k \in \{0, \ldots, N-1\}$, *the following holds:*

$$D_{q_{T-t_k}}^{\delta/N}\left(\widehat{s}_{T-t_k}, s_{T-t_k}\right) \leq \frac{\varepsilon}{\sigma_{T-t_k}} \cdot \frac{1}{\sqrt{T + \log \frac{1}{\gamma}}}$$

*Then, the output distribution* $\widehat{q}_{T-t_N}$ *satisfies*

$$\mathsf{TV}(\widehat{q}_{T-t_N}, q_{T-t_N}) \lesssim \delta + \varepsilon + \mathsf{TV}(Q, Q_{dis}) + \mathsf{TV}(q_T, \mathcal{N}(0, I_d)).$$

*Proof.* Follows by Lemma B.1 and Lemma B.2. $\square$

The next two lemmas from existing works show that the discretization error, $\mathsf{TV}(Q, Q_{\mathrm{dis}})$, is relatively small. Furthermore, as $T$ increases, $q_T$ converges exponentially towards $\mathcal{N}(0, I_d)$.

**Lemma B.4** (Discretization Error, Corollary 1 and eq. (17) in [BBDD24]). *For any* $T \geq 1$, $\gamma < 1$ *and* $N \geq \log(1/\gamma)$, *there exists a sequence of* $N$ *discretization times such that*

$$\mathsf{TV}(Q, Q_{dis}) \lesssim \sqrt{\frac{d}{N}}\left(T + \log \frac{1}{\gamma}\right).$$

**Lemma B.5** (TV between true Gaussian and $q_T$ for large $T$, Proposition 4 in [BBDD24]). *Let* $q$ *be a distribution with a finite second moment of* $m_2^2$. *Then, for* $T \geq 1$ *we have*

$$\mathsf{TV}(q_T, \mathcal{N}(0, I_d)) \lesssim (\sqrt{d} + m_2)e^{-T}.$$

Combining Lemma B.4 and Lemma B.5 with Lemma B.3, we have the following result:

**Corollary B.6.** *Let $q$ be a distribution with finite second moment $m_2^2$. For any $T \geq 1$, $\gamma < 1$ and $N \geq \log(1/\gamma)$, there exists a sequence of discretization times $0 = t_0 < t_1 < \cdots < t_N = T - \gamma$ such that if the following holds for each $k \in \{0, \ldots, N-1\}$:*

$$D_{q_{T-t_k}}^{\delta/N}(\widehat{s}_{T-t_k}, s_{T-t_k}) \leq \frac{\varepsilon}{\sigma_{T-t_k}},$$

*then there exists a sequence of $N$ discretization times such that*

$$\mathsf{TV}(q_\gamma, p_{T-t_N}) \lesssim \delta + \varepsilon\sqrt{T + \log \frac{1}{\gamma}} + \sqrt{\frac{d}{N}}\left(T + \log \frac{1}{\gamma}\right) + (\sqrt{d} + m_2)e^{-T}.$$

This implies our main theorem of this section as a corollary.

**Theorem B.7.** *Let $q$ be a distribution with finite second moment $m_2^2$. For any $\gamma > 0$, there exist $N = \widetilde{O}(\frac{d}{\varepsilon^2 + \delta^2} \log^2 \frac{d + m_2}{\gamma})$ discretization times $0 = t_0 < t_1 < \cdots < t_N < T$ such that if the following holds for every $k \in \{0, \ldots, N-1\}$,*

$$D_{q_{T-t_k}}^{\delta/N}(\widehat{s}_{T-t_k}, s_{T-t_k}) \leq \frac{\varepsilon}{\sigma_{T-t_k}}$$

*then DDPM can produce a sample from a distribution that is within $\widetilde{O}(\delta + \varepsilon\sqrt{\log((d + m_2)/\gamma)})$ in $\mathsf{TV}$ distance of $q_\gamma$ in $N$ steps.*

*Proof.* By setting $T = \log(\frac{\sqrt{d} + m_2}{\varepsilon + \delta})$ and $N = \frac{d(T + \log(1/\gamma))^2}{\varepsilon^2 + \delta^2}$ in Corollary B.6, we have

$$\mathsf{TV}(q_\gamma, p_{T-t_N}) = \widetilde{O}\left(\delta + \varepsilon\sqrt{\log \frac{d + m_2}{\gamma}}\right).$$

$\square$

Furthermore, we present our theorem under the case when $m_2$ lies between $1/\operatorname{poly}(d)$ and $poly(d)$ to provide a clearer illustration.

**Lemma 1.3.** *Let $q$ be a distribution over $\mathbb{R}^d$ with second moment $m_2^2$ between $1/poly(d)$ and $poly(d)$. For any $\gamma > 0$, there exist $N = \widetilde{O}(\frac{d}{\varepsilon^2 + \delta^2} \log^2 \frac{1}{\gamma})$ discretization times $0 = t_0 < t_1 < \cdots < t_N < T$ such that if the following holds for every $k \in \{0, \ldots, N-1\}$:*

$$D_{q_{T-t_k}}^{\delta/N}(\widehat{s}_{T-t_k}, s_{T-t_k}) \leq \frac{\varepsilon}{\sigma_{T-t_k}}$$

*then DDPM can sample from a distribution that is within $\widetilde{O}(\delta + \varepsilon\sqrt{\log(d/\gamma)})$ in $\mathsf{TV}$ distance to $q_\gamma$ in $N$ steps.*

## C  Sample Complexity of Training Diffusion Model

In this section, we present our main theorem that combines our score estimation result in the new (6) sense with prior sampling results (from [BBDD24]) to show that the score can be learned using a number of samples scaling polylogarithmically in $\frac{1}{\gamma}$, where $\gamma$ is the desired sampling accuracy.

**Lemma C.1.** *Let $q_0$ be a distribution with second moment $m_2^2$. Let $\phi_\theta(\cdot)$ be the fully connected neural network with ReLU activations parameterized by $\theta$, with $P$ total parameters and depth $D$. Let $\Theta > 1$. For discretization times $0 = t_0 < t_1 < \cdots < t_N$, let $s_{T-t_k}$ be the true score function of $q_{T-t_k}$. If for each $t_k$, there exists some weight vector $\theta^*$ with $\|\theta^*\|_\infty \leq \Theta$ such that*

$$\mathbb{E}_{x \sim q_{T-t_k}}\left[\|\phi_{\theta^*}(x) - s_{T-t_k}(x)\|_2^2\right] \leq \frac{\delta_{score} \cdot \delta_{train} \cdot \varepsilon^2}{C\sigma_{T-t_k}^2 \cdot N^2} \tag{25}$$

*Then, if we take $m > \widetilde{O}\left(\frac{N(d + \log \frac{1}{\delta_{train}}) \cdot PD}{\varepsilon^2 \delta_{score}} \cdot \log\left(\frac{\max(m_2, 1) \cdot \Theta}{\delta_{train}}\right)\right)$ samples, then with probability $1 - \delta_{train}$, each score $\widehat{s}_{T-t_k}$ learned by score matching satisfies*

$$D_{q_{T-t_k}}^{\delta_{score}/N}(\widehat{s}_{T-t_k}, s_{T-t_k}) \leq \varepsilon/\sigma_{T-t_k}.$$

*Proof.* Note that for each $t_k$, $q_{T-t_k}$ is a $\sigma_{T-t_k}$-smoothed distribution. Therefore, we can use Lemma A.1 by taking $\delta_{\text{train}}/N$ into $\delta_{\text{train}}$ and taking $\delta_{\text{score}}/N$ into $\delta_{\text{score}}$. We have that for each $t_k$, with probability $1 - \delta_{\text{train}}/N$ the following holds:

$$\mathbb{P}_{x \sim q_{T-t_k}} \left[ \|\hat{s}_{T-t_k}(x) - s_{T-t_k}(x)\| \le \varepsilon/\sigma_{T-t_k} \right] \ge 1 - \delta_{\text{score}}/N.$$

By a union bound over all the steps, we conclude the proposed statement. $\square$

Now we present our main theorem.

**Theorem C.2** (Main Theorem, Full Version). *Let $q$ be a distribution of $\mathbb{R}^d$ with second moment $m_2^2$. Let $\phi_\theta(\cdot)$ be the fully connected neural network with ReLU activations parameterized by $\theta$, with $P$ total parameters and depth $D$. Let $\Theta > 1$. For any $\gamma > 0$, there exist $N = \widetilde{O}(\frac{d}{\varepsilon^2+\delta^2} \log^2 \frac{m_2+1/m_2}{\gamma})$ discretization times $0 = t_0 < \cdots < t_N < T$ such that if for each $t_k$, there exists some weight vector $\theta^*$ with $\|\theta^*\|_\infty \le \Theta$ such that*

$$\mathop{\mathbb{E}}_{x \sim q_{T-t_k}} \left[ \|\phi_{\theta^*}(x) - s_{T-t_k}(x)\|_2^2 \right] \le \frac{\delta \cdot \varepsilon^3}{CN^2\sigma_{T-t_k}^2} \cdot \frac{1}{\log \frac{d+m_2+1/m_2}{\gamma}}$$

*for sufficiently large constant $C$, then consider the score functions trained from*

$$m > \widetilde{O} \left( \frac{N(d + \log \frac{1}{\delta}) \cdot PD}{\varepsilon'^3} \cdot \log \left( \frac{\max(m_2, 1) \cdot \Theta}{\delta} \right) \cdot \log \frac{m_2 + 1/m_2}{\gamma} \right)$$

*i.i.d. samples of $q$, with $1 - \delta$ probability, DDPM can sample from a distribution $\varepsilon$-close in $\mathsf{TV}$ to a distribution $\gamma m_2$-close in 2-Wasserstein to $q$ in $N$ steps.*

*Proof.* Note that for an arbitrary $t > 0$, the 2-Wasserstein distance between $q$ and $q_t$ is bounded by $O(tm_2 + \sqrt{td})$. Therefore, by choosing $t_N = T - \min(\gamma, \gamma^2 m_2^2/d)$, Theorem B.7 shows that by choosing $N = \widetilde{O}(\frac{d}{\varepsilon'^2+\delta^2} \log^2 \frac{d+m_2}{\min(\gamma, \gamma^2 m_2^2/d)})$, we only need

$$D_{q_{T-t_k}}^{\varepsilon/N} (\hat{s}_{T-t_k}, s_{T-t_k}) \le \frac{\varepsilon'}{\sigma_{T-t_k} \sqrt{\log \frac{d+m_2}{\min(\gamma, \gamma^2 m_2^2/d)}}}$$

then DDPM can produce a sample from a distribution within $\widetilde{O}(\varepsilon')$ in $\mathsf{TV}$ distance to a distribution $\gamma m_2$-close in 2-Wasserstein to $q$ in $N$ steps. Note that

$$\log \frac{d + m_2}{\min(\gamma, \gamma^2 m_2^2/d)} \lesssim \log \frac{d + m_2 + 1/m_2}{\gamma}.$$

Therefore, we only need to take $\widetilde{O}(\frac{d}{\varepsilon'^2+\delta^2} \log^2 \frac{m_2+1/m_2}{\gamma})$ steps. Therefore, to achieve this, we set $\delta_{\text{train}} = \delta$, $\delta_{\text{score}} = \varepsilon'$, and $\varepsilon = \varepsilon'/\sqrt{\log \frac{d+m_2+1/m_2}{\gamma}} \lesssim \varepsilon'/\sqrt{\log \frac{d+m_2}{\min(\gamma, \gamma^2 m_2^2/d)}}$ in Lemma C.1. This gives us the result that with

$$m > \widetilde{O} \left( \frac{N(d + \log \frac{1}{\delta}) \cdot PD}{\varepsilon'^3} \cdot \log \left( \frac{\max(m_2, 1) \cdot \Theta}{\delta} \right) \cdot \log \frac{m_2 + 1/m_2}{\gamma} \right)$$

samples, we can satisfy the score requirement given the assumption in the statement. $\square$

For cleaner statement, we present this theorem under the case when $m_2$ lies between $1/\operatorname{poly}(d)$ and $\operatorname{poly}(d)$ and achieving training success probability of 99%. This gives the quantitative version of Theorem 1.2.

**Theorem C.3** (Main Theorem, Quantitative Version). *Let $q$ be a distribution of $\mathbb{R}^d$ with second moment $m_2^2$. Let $\phi_\theta(\cdot)$ be the fully connected neural network with ReLU activations parameterized by $\theta$, with $P$ total parameters and depth $D$. Let $\Theta > 1$. For any $\gamma > 0$, there exist $N = \widetilde{O}(d \log^2 \frac{1}{\gamma})$ discretization times $0 = t_0 < \cdots < t_N < T$ such that if for each $t_k$, there exists some weight vector $\theta^*$ with $\|\theta^*\|_\infty \le \Theta$ such that*

$$\mathop{\mathbb{E}}_{x \sim q_{T-t_k}} \left[ \|\phi_{\theta^*}(x) - s_{T-t_k}(x)\|_2^2 \right] \le \frac{\delta \cdot \varepsilon^3}{CN^2\sigma_{T-t_k}^2 \log \frac{d}{\gamma}}$$

*for a sufficiently large constant $C$, then consider the score functions trained from*

$$m \geq \widetilde{O}\left(\frac{d^2 PD}{\varepsilon^3} \cdot \log \Theta \cdot \log^3 \frac{1}{\gamma}\right)$$

*i.i.d. samples of $q$. With $99\%$ probability, DDPM using these score functions can sample from a distribution $\varepsilon$-close in $\mathsf{TV}$ to a distribution $\gamma m_2$-close in 2-Wasserstein to $q$ in $N$ steps.*

## D  Hardness of Learning in $L^2$

In this section, we give proofs of the hardness of the examples we mention in Section 4.

**Lemma 4.1.** *Let $R$ be sufficiently larger than $\sigma$. Let $p_1$ be the distribution $(1 - \eta)\mathcal{N}(0, \sigma^2) + \eta\mathcal{N}(-R, \sigma^2)$ with corresponding score function $s_1$, and let $p_2$ be $(1 - \eta)\mathcal{N}(0, \sigma^2) + \eta\mathcal{N}(R, \sigma^2)$ with score $s_2$, such that $\eta = \frac{\varepsilon^2 \sigma^2}{R^2}$. Then, given $m < \frac{R^2}{\varepsilon^2 \sigma^2}$ samples from either distribution, it is impossible to distinguish between $p_1$ and $p_2$ with probability larger than $1/2 + o_m(1)$. But,*

$$\mathop{\mathbb{E}}_{x \sim p_1}\left[\|s_1(x) - s_2(x)\|^2\right] \gtrsim \frac{\varepsilon^2}{\sigma^2} \qquad and \qquad \mathop{\mathbb{E}}_{x \sim p_2}\left[\|s_1(x) - s_2(x)\|^2\right] \gtrsim \frac{\varepsilon^2}{\sigma^2}.$$

*Proof.*

$$\mathsf{TV}(p_1, p_2) \gtrsim \eta$$

So, it is impossible to distinguish between $p_1$ and $p_2$ with fewer than $O\left(\frac{1}{\eta}\right)$ samples with probability $1/2 + o_m(1)$.

The score $L^2$ bound follows from calculation. $\qquad\square$

**Lemma 4.2.** *Let $S$ be sufficiently large. Consider the distribution $\widehat{p} = \eta\mathcal{N}(0, \sigma^2) + (1 - \eta)\mathcal{N}(S, \sigma^2)$ for $\eta = \frac{Se^{-\frac{S^2}{2} + 10\sqrt{\log m} \cdot S}}{10\sqrt{\log m}}$, and let $\widehat{s}$ be its score function. Given $m$ samples from the standard Gaussian $p^* = \mathcal{N}(0, \sigma^2)$ with score function $s^*$, with probability at least $1 - \frac{1}{poly(m)}$,*

$$\widehat{\mathbb{E}}\left[\|\widehat{s}(x) - s^*(x)\|^2\right] \leq \frac{1}{\sigma^2} e^{-O(S\sqrt{\log m})} \quad but \quad \mathop{\mathbb{E}}_{x \sim p^*}\left[\|\widehat{s}(x) - s^*(x)\|^2\right] \gtrsim \frac{S^2}{m\sigma^4}.$$

*Proof.* Let $X_1, \ldots, X_m \sim p^*$ be the $m$ samples from $\mathcal{N}(0, \sigma^2)$. With probability at least $1 - \frac{1}{poly(m)}$, every $X_i \leq 2\sigma\sqrt{\log m}$. Now, the score function of the mixture $\widehat{p}$ is given by

$$\widehat{s}(x) = \frac{-\frac{x}{\sigma^2} - \frac{x-S}{\sigma^2}\left(\frac{1-\eta}{\eta}\right)e^{-\frac{S^2}{2} + Sx}}{1 + \left(\frac{1-\eta}{\eta}\right)e^{-\frac{S^2}{2} + Sx}}$$

For $x \leq 2\sqrt{\log m}$,

$$\widehat{s}(x) = -\frac{x}{\sigma^2}\left(1 + \frac{e^{-O(S\sqrt{\log m})}}{S}\right) + \frac{1}{\sigma^2}e^{-O(S\sqrt{\log m})}$$

So,

$$\widehat{\mathbb{E}}\left[\|\widehat{s}(x) - s^*(x)\|^2\right] \leq \frac{1}{\sigma^2} e^{-O(S\sqrt{\log m})}$$

On the other hand,

$$\mathop{\mathbb{E}}_{x \sim p^*}\left[\|\widehat{s}(x) - s^*(x)\|^2\right] \gtrsim \frac{S^2}{m\sigma^4}$$

$\qquad\square$

# E   Discussion of [BMR20]

Here, we present a brief, self-contained discussion of the prior work [BMR20]. The main result of that work on score estimation is as follows.

**Theorem E.1** (Proposition 9 of [BMR20], Restated). *Let $\mathcal{F}$ be a class of $\mathbb{R}^d$-valued functions, all of with are $\frac{L}{2}$-Lipschitz, with values supported on the Euclidean ball with radius $R$, and containing uniformly good approximations of the true score $s_t$ on this ball. Given $n$ samples from $q_t$, if we let $\widehat{s}_t$ be the empirical minimizer of the score-matching loss, then with probability at least $1 - \delta$,*

$$\mathbb{E}_{x \sim q_t}\left[\|\widehat{s}_t(x) - s_t(x)\|^2\right] \lesssim (LR + B)^2 \left(\log^2 n \cdot \mathcal{R}_n^2(\mathcal{F}) + \frac{d\left(\log\log n + \log\frac{1}{\delta}\right)}{n}\right)$$

*Here, $B$ is a bound on $\|s_t(0)\|$, and $\mathcal{R}_n(\mathcal{F})$ is the* Rademacher complexity *of $\mathcal{F}$ over $n$ samples.*

In the setting we consider, for the class of neural networks with weight vector $\theta$ such that $\|\theta\|_\infty \leq \Theta$ with $P$ parameters and depth $D$, it was shown in [Sel24] that

$$\mathcal{R}_n(\mathcal{F}) \lesssim \frac{R\sqrt{d}}{\sqrt{n}}(\Theta\sqrt{P})^D \cdot \sqrt{D}$$

Moreover, the true score is $\frac{R}{\sigma_t}$ Lipschitz, so that if we further restrict ourselves to neural networks have the same Lipschitzness, $L \asymp \frac{R}{\sigma_t}$. $B$ can be bounded by $\frac{\sqrt{d}}{\sigma_t}$.

Thus, an application of the theorem to our setting gives a sample complexity bound of $\widetilde{O}\left(\frac{R^6 d + R^2 d^2}{\varepsilon^2} \cdot (\Theta^2 P)^D \sqrt{D} \log\frac{1}{\delta}\right)$ for squared $L^2$ score error $O\left(\frac{\varepsilon^2}{\sigma_t^2}\right)$. In order to support $\sigma_t$-smoothed scores, we need to set $R \gtrsim \frac{\sqrt{d}}{\sigma_t}$, for a sample complexity of $\widetilde{O}\left(\frac{d^{5/2} R^3}{\sigma_t^3 \varepsilon^2}(\Theta^2 P)^D \sqrt{D} \log\frac{1}{\delta}\right)$ to learn an approximation to $s_t$ with squared $L^2$ error $O\left(\frac{\varepsilon^2}{\sigma_t^2}\right)$.

So, we obtain the following corollary.

**Corollary E.2.** *Let $q$ be a distribution over $\mathbb{R}^d$ supported over the Euclidean ball with radius $R \leq O\left(\sqrt{d}/\sigma_t\right)$. Let $\mathcal{F}$ be the set of neural networks $\phi_\theta(\cdot)$, with weight vector $\theta$, $P$ parameters and depth $D$, with $\|\theta\|_\infty \leq \Theta$ and values supported on the ball of radius $R$, and that are $O(R/\sigma_t)$-Lipschitz. Suppose $\mathcal{F}$ contains a network with weights $\theta^*$ such that $\phi_{\theta^*}$ provides uniformly good approximation of the true score $s_t$ over the ball of radius $R$. Given $n$ samples from $q_t$, if we let $\widehat{s}_t$ be the empirical minimizer of the score-matching loss, then with probability $1 - \delta$,*

$$\mathbb{E}_{x \sim q_t}\left[\|\widehat{s}_t(x) - \phi_{\theta^*}(x)\|^2\right] \leq \widetilde{O}\left(\frac{\varepsilon^2}{\sigma^2}\right)$$

*for $n > \widetilde{O}\left(\frac{d^{5/2} R^3}{\sigma_t^3 \varepsilon^2}\left(\Theta^2 P\right)^D \sqrt{D} \log\frac{1}{\delta}\right)$.*

To learn score approximations in $L^2$ for all the $N$ relevant timesteps in order to achieve TV error $\varepsilon$ and Wasserstein error $\gamma \cdot R$, this implies a sample complexity bound of $\widetilde{O}\left(\frac{d^{5/2}}{\gamma^3 \varepsilon^2}(\Theta^2 P)^D \sqrt{D} \log\frac{N}{\delta}\right) = \widetilde{O}\left(\frac{d^{5/2}}{\gamma^3 \varepsilon^2}(\Theta^2 P)^D \sqrt{D} \log\frac{1}{\delta}\right)$ for our final choice of $N$.

# F   Utility Results

**Lemma F.1** (From [GLP23]). *Let $f$ be an arbitrary distribution on $\mathbb{R}^d$, and let $f_\Sigma$ be the $\Sigma$-smoothed version of $f$. That is, $f_\Sigma(x) = \mathbb{E}_{y \sim f}\left[(2\pi)^{-d/2}\det(\Sigma)^{-1/2}\exp\left(-\frac{1}{2}(x-Y)^T\Sigma^{-1}(x-Y)\right)\right]$. Let $s_\Sigma$ be the score function of $f_\Sigma$. Let $(X, Y, Z_\Sigma)$ be the joint distribution such that $Y \sim f$, $Z_\Sigma \sim \mathcal{N}(0, \Sigma)$ are independent, and $X = Y + Z_\Sigma \sim f_\Sigma$. We have for $\varepsilon \in \mathbb{R}^d$,*

$$\frac{f_\Sigma(x+\varepsilon)}{f_\Sigma(x)} = \mathbb{E}_{Z_\Sigma|x}\left[e^{-\varepsilon^T\Sigma^{-1}Z_\Sigma - \frac{1}{2}\varepsilon^T\Sigma^{-1}\varepsilon}\right]$$

*so that*

$$s_\Sigma(x) = \mathbb{E}_{Z_\Sigma|x}\left[-\Sigma^{-1}Z_\Sigma\right]$$

**Lemma F.2** (From [HKZ12], restated). *Let $x$ be a mean-zero random vector in $\mathbb{R}^d$ that is $\Sigma$-subgaussian. That is, for every vector $v$,*

$$\mathbb{E}\left[e^{\lambda\langle x,v\rangle}\right] \le e^{\lambda^2 v^T \Sigma v/2}$$

*Then, with probability $1 - \delta$,*

$$\|x\| \lesssim \sqrt{\text{Tr}(\Sigma)} + \sqrt{2\|\Sigma\|\log\frac{1}{\delta}}$$

**Lemma F.3.** *For $s_\Sigma$ the score function of an $\Sigma$-smoothed distribution where $\Sigma = \sigma^2 I$, we have that $v^T s_\Sigma(x)$ is $O(1/\sigma^2)$-subgaussian, when $x \sim f_\Sigma$ and $\|v\| = 1$.*

*Proof.* We have by Lemma F.1 that

$$s_\Sigma(x) = \mathbb{E}_{Z_\Sigma|x}\left[\Sigma^{-1} Z_\Sigma\right]$$

So,

$$\begin{aligned}
\mathbb{E}_{x\sim f_\Sigma}\left[(v^T s_\Sigma(x))^k\right] &= \mathbb{E}_{x\sim f_\Sigma}\left[\mathbb{E}_{Z_\Sigma|x}\left[v^T \Sigma^{-1} Z_\Sigma\right]^k\right] \\
&\le \mathbb{E}_{Z_\Sigma}\left[(v^T\Sigma^{-1} Z_\Sigma)^k\right] \\
&\le \frac{k^{k/2}}{\sigma^k} \quad \text{since } v^T Z_\Sigma \sim \mathcal{N}(0,\sigma^2)
\end{aligned}$$

The claim follows. $\square$

**Lemma F.4.** *Let $\Sigma = \sigma^2 I$, and let $x \sim f_\Sigma$. We have that with probability $1 - \delta$,*

$$\|s_\Sigma(x)\|^2 \lesssim \frac{d + \log\frac{1}{\delta}}{\sigma^2}$$

*Proof.* Follows from Lemmas F.3 and F.2. $\square$

**Lemma F.5.** *For $z \sim \mathcal{N}(0,\sigma^2 I_d)$, with probability $1 - \delta$,*

$$\left\|\frac{z}{\sigma^2}\right\| \lesssim \frac{\sqrt{d + \log\frac{1}{\delta}}}{\sigma}$$

*Proof.* Note that $\|z\|^2$ is chi-square, so that we have for any $0 \le \lambda < \sigma^2/2$,

$$\mathbb{E}_z\left[e^{\lambda\|\frac{z}{\sigma^2}\|^2}\right] \le \frac{1}{(1 - 2(\lambda/\sigma^2))^{d/2}}$$

The claim then follows by the Chernoff bound. $\square$

**Lemma F.6.** *For $x \sim f_\Sigma$, with probability $1 - \delta$,*

$$\mathbb{E}_{Z_\Sigma|x}\left[\frac{\|Z_\Sigma\|}{\sigma^2}\right] \lesssim \frac{\sqrt{d + \log\frac{1}{\delta}}}{\sigma}$$

*Proof.* Since $Z_\Sigma \sim \mathcal{N}(0,\sigma^2 I_d)$ so that $\|Z_\Sigma\|^2$ is chi-square, we have that for any $0 \le \lambda < \sigma^2/2$, by Jensen's inequality,

$$\mathbb{E}_{x\sim f_\Sigma}\left[e^{\lambda\mathbb{E}_{Z_\Sigma|x}\left[\frac{\|Z_\Sigma\|}{\sigma^2}\right]^2}\right] \le \mathbb{E}_{Z_\Sigma}\left[e^{\lambda\frac{\|Z_\Sigma\|^2}{\sigma^4}}\right] \le \frac{1}{(1 - 2(\lambda/\sigma^2))^{d/2}}$$

The claim then follows by the Chernoff bound. That is, setting $\lambda = \sigma^2/4$, for any $t > 0$,

$$\mathbb{P}_{x\sim f_\Sigma}\left[\mathbb{E}_{Z_\Sigma|x}\left[\frac{\|Z_\Sigma\|}{\sigma^2}\right]^2 \ge t\right] \le \frac{\mathbb{E}_{x\sim f_\Sigma}\left[e^{\lambda\mathbb{E}_{Z_\Sigma|x}\left[\frac{\|Z_\Sigma\|}{\sigma^2}\right]^2}\right]}{e^{\lambda t}} \le 2^{d/2}e^{-t\sigma^2/4} = 2^{\frac{d\ln 2}{2} - \frac{t\sigma^2}{4}}$$

For $t = O\left(\frac{d+\log\frac{1}{\delta}}{\sigma^2}\right)$, this is less than $\delta$. $\square$

**Lemma F.7** ([KN])**.** *Let $X$ be a non-negative random variable such that for every $t \geq 0$, $\mathbb{P}[X \geq t] \leq \exp(-\lambda t)$ for some constant $\lambda \geq 0$. Then, for $K \geq 0$,*

$$\mathbb{E}[X|X \geq K] \lesssim \mathbb{E}[X] + \frac{1}{\lambda} \log \left( \frac{1}{\mathbb{P}[X \geq K]} \right)$$

*Proof.* Let $p := \mathbb{P}[X \geq K]$ and denote the distribution of $X$ as $\mathcal{P}$. Consider the sequence of iid samples $\{X_i\}_{i \geq 0}$ sampled from $\mathcal{P}$. Let $T$ be the first index $i$, where $X_i \geq K$. Then the distribution of $X_T$ is the conditional distribution of $X$ given $X \geq K$. For any $m > 0$,

$$X_T \leq \sum_{k \geq 1} \mathbb{1}\left((k-1)m < T \leq km\right) \sup_{(k-1)m < i \leq km} X_i$$
$$\leq \sum_{k \geq 1} \mathbb{1}\left((k-1)m < T\right) \sup_{(k-1)m < i \leq km} X_i$$

Now, $\mathbb{1}\left((k-1)m < T\right)$ and $\sup_{(k-1)m < i \leq km} X_i$ are independent since $\mathbb{1}\left((k-1)m < T\right)$ depends only on $\{X_i\}_{i \in [1,(k-1)m]}$. Thus,

$$\mathbb{E}[X|X \geq K] = \mathbb{E}[X_T] \leq \sum_{k \geq 1} \mathbb{E}\left[\mathbb{1}\left((k-1)m < T\right)\right] \mathbb{E}\left[ \sup_{(k-1)m < i \leq km} X_i \right]$$
$$\lesssim (1-p)^{(k-1)m} \left( \mathbb{E}[X] + \frac{\log m}{\lambda} \right)$$
$$\lesssim \frac{1}{1 - (1-p)^m} \cdot \left( \mathbb{E}[X] + \frac{\log m}{\lambda} \right)$$

Choosing $m = \frac{1}{p}$ so that $(1-p)^m \leq \frac{1}{e}$ gives the claim. $\qquad\square$

**Theorem F.8** (Girsanov's theorem)**.** *For $t \in [0,T]$, let $\mathcal{L}_t = \int_0^t b_s \, \mathrm{d}B_s$ where $B$ is a $Q$-Brownian motion. Assume Novikov's condition is satisfied:*

$$\mathbb{E}_Q \left[ \exp \left( \frac{1}{2} \int_0^T \|b_s\|_2^2 \, \mathrm{d}s \right) \right] < \infty.$$

*Then*

$$\mathcal{E}(\mathcal{L})_t := \exp \left( \int_0^t b_s \, \mathrm{d}B_s - \frac{1}{2} \int_0^t \|b_s\|_2^2 \, \mathrm{d}s \right)$$

*is a $Q$-martingale and*

$$\widetilde{B}_t := B_t - \int_0^t b_s \, \mathrm{d}s$$

*is a Brownian motion under $P$ where $P := \mathcal{E}(\mathcal{L})_T Q$, the probability distribution with density $\mathcal{E}(\mathcal{L})_T$ w.r.t. $Q$.*

