# OpenReview forum: "Improved Sample Complexity Bounds for Diffusion Model Training"
_NeurIPS.cc/2024/Conference — NeurIPS 2024 poster_

### Official Review · Reviewer_acEK · 2024-07-08

**Soundness:** 3
**Presentation:** 3
**Contribution:** 2
**Rating:** 5
**Confidence:** 2

**Summary:**

This paper investigates theoretical guarantees for the performance of diffusion models. More precisely, while the previous works have mainly focused on the iteration complexity by assuming to have access to an accurate diffusion model, this paper targets the question of the sample complexity for learning an accurate diffusion model (score function). The main contribution is an exponential improvement of the existing bound with respect to Wasserstein error and depth.

**Strengths:**

The main contribution is to improve the existing sample complexity for learning the score function, exponentially with respect to some on $\Theta$, $\gamma$, and $D$. This may contribute to a better theoretical understanding of diffusion-based models.

**Weaknesses:**

- The impact of the results obtained and the scope of the theoretical insights are limited. I am not sure that the results of this paper bring concrete new implications/understandings for diffusion models.

- Some parts of the paper are not clear enough (see Questions section)

**Questions:**

1. Can the authors develop more why and when the dependence on $\theta$, $\gamma$, $D$ and $P$ is more important than the dependence on $\epsilon$? Can a numerical estimate be provided here?


2. Can the authors discuss assumption A2 in more detail? Is it true that the assumption implies $\inf_{f} E_{X \sim q_t} [||f(x) - s_t(x)||_2]=0$? Does this assumption implicitly assume that the score function is "simple" enough to belong to the family $\mathcal{F}$? Does it lead to any limitation in practice?

3. What are $\gamma$ and $R$ in line 41? They seem to be undefined up to that point.

4.  For clarity, it may be better to replace $z_i$ with $z_{i,t}$ in line 54 to show that such samples are distinct and i.i.d. for each sample and time index.

5. The paper and the considered setting 1.1. depend strongly on the DDPM algorithm. To make the paper self-contained, it is necessary to describe this algorithm.

6. In Setting 1.1, does $P$ denote the number of parameters per layer?

**Limitations:**

Not Applicable.

---

> ### Author Rebuttal · Authors · 2024-08-07
>
> We appreciate your thorough review and valuable suggestions on the presentation of the paper. We appreciate the time and effort you have invested in providing this feedback.
>
> Implications of our work
> ----
>
> Our work is the first to show a polynomial sample complexity for learning a diffusion models using a deep network under a reasonable assumption -- that the scores of the distribution can be represented using
> a $P$-parameter neural network. From a theoretical point of view, we believe this is a significant contribution -- it was not clear prior to our work whether minimizing the score-matching objective is sufficient to learn a diffusion model using a small number of samples.
>
> More broadly, we hope that this style of assumption will inspire future work in understanding diffusion models and other deep learning models.
> %We would like to highlight the concrete new insights our research contributes to the field of diffusion models.
>
>
> Responses to specific questions
> ----
>
> > *Can the authors develop more why and when the dependence on $\theta$, $\gamma$, $D$ and $P$ is more important than the dependence on $\varepsilon$? Can a numerical estimate be provided here?*
>
> To illustrate the dependence on these parameters, we provide some natural choices of parameters:
>
> - TV error $\varepsilon = 0.01$.
> - Wasserstein error $\gamma = 0.01$.
> - Space dimension $d = 12288 = 64 \times 64 \times 3$.
> - Total Parameter Number $P = 270M = 2.7 \times 10^8$.
> - Network Depth $D = 12$.
> - Parameter Weight Range $\Theta = 5$.
>
> For parameter number and the space dimension, we used the parameters in stable diffusion[1]. Our bound gives $\frac{d^2 PD}{\varepsilon^3} \cdot \log \Theta \cdot \log^3 \left(\frac{1}{\gamma}\right) \approx 7.7 \times 10^{25}$, compared to the previous bound of $\frac{d^{5/2}}{\gamma^3 \varepsilon^2} \left(\Theta^2 P\right)^D \sqrt{D} \approx 5.2 \times 10^{138}$.
>
> While we do not claim that $\theta$, $\gamma$, $D$, and $P$ are universally more important than $\varepsilon$, our results demonstrate a significant *exponential* improvement in $D$ and $\gamma$ with only a minor loss in $\varepsilon$, resulting in a substantially better overall bound.
>
> > *Is it true that assumption A2 implies $\inf_f \mathbb{E}_{X \sim q_t}[\| f(x) - s_t(x) \|_2] = 0$? Does this assumption implicitly assume that the score function is "simple" enough to belong to the family?*
>
> Actually, we only need the $L^2$ error to be polynomially small, namely
> $\frac{\delta \cdot \varepsilon^3}{N^2 \sigma^2_{T - t_k} \log \frac{d}{\gamma}}$,
> as specified in Theorem C.3. We will put this in the main body of the paper.  Intuitively, this assumption states that neural networks can effectively represent the score function. This assumption is
> both reasonable (neural networks have proven to be surprisingly
> powerful function approximators) and necessary (if the distribution's
> score isn't well approximated by a neural network, diffusion won't
> work).
>
> > *What are $\gamma$  and $R$ in line 41?*
>
> In line 41, $R$ represents the radius of the support size of the distribution, and $\gamma$ is a parameter used to specify the Wasserstein error.
>
> > *In Setting 1.1, does $P$ denote the number of parameters per layer?*
>
> In Setting 1.1, $P$ denotes the total number of parameters, not just per layer. We will clarify this in the final version to avoid any ambiguity.
>
> [1] Rombach, Robin, et al. "High-resolution image synthesis with latent diffusion models." Proceedings of the IEEE/CVF conference on computer vision and pattern recognition. 2022.

---

> > ### Comment · Reviewer_acEK · 2024-08-09
> >
> > Thanks for the response and the numerical estimate. Apart from the presentation, which I believe will be improved by the authors, I think the main debate is about the importance of the paper, as I mentioned in the first weakness and discussed in more detail by other reviewers. Unfortunately, I am not in a position to make an informed judgment, but I think the paper contains interesting results and I maintain my rating.

---

### Official Review · Reviewer_r7NG · 2024-07-11

**Soundness:** 3
**Presentation:** 3
**Contribution:** 3
**Rating:** 7
**Confidence:** 3

**Summary:**

In this paper, the authors analyze the sample complexity of training a score-based diffusion model. They show that, with a sufficiently expressive neural network,  \tilde O(d^2 P D log \Theta \log^3 \frac{1}{\gamma} / \epsilon^3) samples are needed to learn an accurate diffusion model. Compared to the existing result, their bound has exponentially better dependence on \Theta, \gamma and D, and a better polynomial dependence on d and P, at the cost of a worse polynomial dependence in \epsilon.

**Strengths:**

The paper improves the existing bound for training a diffusion model with sufficiently expressive neural network. They analyze the barrier for L_2 accuracy as a criterion and propose to use a (1-\delta)-quantile error in their analysis, which would finally scale polylogarithmically in 1/\gamma, and suffices for fast sampling via the reverse SDE. Basically, the organization of this paper is good, and the proof sketch is clear, though it is a hard to check all the details.

**Weaknesses:**

1. Generally, this paper discusses the training sample complexity of a score-based diffusion model. The assumption is somehow strong, e.g., the neural networks can represent the score. Besides, the author do not include the optimization error in the analyses.

**Questions:**

1. As the bound in lemma 1.4 is with probability and in lemma 1.3, the requirement must hold for all N,  what if we make sampling step N \to \infty in Theorem 1.2? Does the bound in lemma 4 for each t independent or is controlled by the same stochasticity?

2. What is the quantitive requirement on \mathcal{F} in Assumption 2? What does the "sufficiently small" mean?

3.Why is it required to constraint the functions represented by a fully connected neural network with ReLU activations and depth D, with P parameters, each bounded by \Theta? Is it possible to be extended to other function classes?

**Limitations:**

Not applicable.

---

> ### Author Rebuttal · Authors · 2024-08-07
>
> We appreciate your comments and are glad that you like and support the paper.
>
> **The assumption that networks can
>   represent the score is somehow strong.**
> We agree that it is somewhat strong, but view this more as a statement
> about data; analogous to "the data is sparse in X basis".  Over the
> past decade neural networks have proven to be surprisingly effective
> function approximators.  We ask: supposing this is true, how many
> samples do you need to accurately learn a generative model?
>
> (And if it's not true, then trying to learn the score with a neural network is doomed to fail regardless of how many samples you use.)
>
> **Q1 (Applying Lemma 1.4  for all N):** That's right, Theorem 1.2 applies Lemma 1.4 in a union bound over
> the N time steps to invoke Lemma 1.3.  Since the dependence on
> probability $\delta_{train}$ is logarithmic, this loses at most a
> $\log N$ factor.  Not good if $N$ is really $\infty$, but fine for more
> reasonable values of $N$.
>
> That said, there is an interesting question here.  In our analysis, it
> doesn't matter if the image samples are chosen independently or
> jointly across different scales.  In practice, they are typically
> chosen jointly (so each sample image is then used at all noise
> scales).  It seems plausible to us that jointly sampling images could
> improve the sample complexity in Theorem 1.2 from $d^2$ down to
> $d$, by correlating the "bad" events across scales.
>
> **Q2 (Quantitative Assumption 2):** We give the precise bound in Theorem C.3 in the appendix.  It's a
> polynomial bound, namely $\frac{\delta \varepsilon^3}{N^2 \sigma^2
> \log \frac{d}{\gamma}}$.
>
>
> **Q3 (Other function classes):** Our proof actually applies to any Lipschitz activation function,
> we should have specified this.  We think measuring a neural network by
> its parameter count / weight / depth is a simple way to bound its
> complexity.  One can certainly analyze other classes -- Lemma A.2
> applies to arbitrary finite function classes, and one can take a net
> for other classes -- but we're not sure of any more general statement
> that applies cleanly to neural networks.
>
> (Some work (e.g. BMR20) tries to use Rademacher complexity to get a
> general statement, but the Rademacher complexity of neural networks is
> exponential in their depth so this is unsatisfying.)

---

> > ### Comment · Reviewer_r7NG · 2024-08-12
> >
> > Thank you for your reply!
> >
> > 1. I am still doubt about the assumption that the score function can be effectively. As the authors constraint the functions represented by a fully connected neural network with ReLU activations and depth D, with P parameters, each bounded by \Theta. The question now becomes whether this family is strong enough to represent the score of the complex dynamics? What is the influence of these network parameters other than the representative ability?
> >
> > 2. As confirmed by the authors, I am a little confused. If N could not be too large, so how do we bound the discretizaiton error?  The authors should discuss this more sufficiently, and especially when jump from the individual bound to the union bound.
> >
> > 3. Note that the authors did not discuss the optimization error, it is really strange to use the description "trained from". Does the authors assume the global minimum can be achieved?

---

> > > ### Author Response · Authors · 2024-08-14
> > >
> > > Thank you for your reply! We respond to your questions below:
> > >
> > > 1. Our goal is to understand the sample complexity of using the score matching objective to approximate the score by a neural network for diffusion, because that is the process people are using successfully in practice.  There are a couple questions you might be asking, and we're not sure which:
> > >     - Why study this particular family of neural networks (relu, fully connected, etc.)?  Our techniques are pretty general, and could be applied to any continuous neural network with Lipschitz activation.  This just seemed like the simplest setting to describe (of course we need some bound on the complexity: bigger, more complex networks need more samples).
> > >     - Why would we expect neural networks to be able to represent the score accurately?  We think this is somewhat orthogonal to the sample complexity question: if they can't represent the score, then diffusion isn't going to work no matter how many samples you have.  In practice, neural networks do seem to be able to represent scores accurately.  Why that happens is a very interesting question, but we view it as more a question about *data* than about neural networks or optimization.  (Analogous to: images tend to be sparse in the wavelet basis.)  It's trivial to construct synthetic data that cannot be represented by a neural network (or any other compact representation).
> > >
> > > 2. **Choice of $N$**.
> > > There are competing factors: $N$ needs to be large enough for the discretization error to be small, but the sample complexity and sampling time degrade with increasing $N$.  As we specify in Lemma 1.3, the sweet spot is $N = \tilde{O}(\frac{d}{\varepsilon^2 + \delta^2} \log^2 \frac{1}{\gamma})$.
> > >
> > > 3. **Optimization error.**
> > > We did assume that the global minimum can be achieved, as specified in Setting 1.1. This is a standard setting in the line of works that aim to bound sample complexity. While analyzing the optimization process is indeed interesting, it is also extremely challenging. Our focus is on determining how many samples are needed for the minimizer of the neural network to be effective in the diffusion process.
> > >
> > >     In practice, SGD has been shown to approximate the global minimum well. Our analysis can easily extend to handle approximation error in the optimization. Given an approximate minimizer with error comparable to the bound we give on line 712, the same analysis would give the same result.
> > >
> > >     We can clarify the wording "trained from" to specify that we mean the ERM of the score matching objective on the given samples.

---

### Official Review · Reviewer_6Nj2 · 2024-07-12

**Soundness:** 3
**Presentation:** 2
**Contribution:** 3
**Rating:** 5
**Confidence:** 3

**Summary:**

The paper studies the sample complexity of training diffusion models. The bound derived in the paper is exponentially better than the previous results. The paper also discusses the difficulty of learning the score function in $L^2$.

**Strengths:**

1. The paper derives better sample complexity results for diffusion models. The bound in this work is exponentially better than the existing results.
2. The paper discusses the difficulties of learning the score in $L^2$. Two examples are provided to show this challenge.

**Weaknesses:**

1. The major concern is the presentation of this paper. I understand the results are important. However, the authors should clearly position this work in the literature. I suggest the authors provide a table that compares the sample complexity to the ones in prior work. In particular, the authors should discuss the improvement in terms of both the TV bound and the Wasserstein results.
2. I notice the authors use BMR20 as a benchmark in Section 1.1. However, there are many better sample complexity results on diffusion models since that work. I suggest the authors carefully state how this work differs from those results. Also, I suggest the authors emphasize the main technical contributions in the proof instead of stating all the steps one by one. In particular, which step and what technique lead to the exponential improvement?
3. The importance of Section 4 is not clear. Recent work has explored the learning score function in $L^2$; see [1-5]. All the work listed shows learning the score in $L^2$ is possible, and some of them show the minimax optimal rate. Specifically, learning the score function for an arbitrary distribution can be hard. However, learning the denoised score function during the forward process can be much easier. Therefore, I suggest the authors clarify why the examples in Section 4 can be interesting.

[1] Chen, Minshuo, et al. "Score approximation, estimation and distribution recovery of diffusion models on low-dimensional data." International Conference on Machine Learning. PMLR, 2023.

[2] Han, Yinbin, Meisam Razaviyayn, and Renyuan Xu. "Neural network-based score estimation in diffusion models: Optimization and generalization." arXiv preprint arXiv:2401.15604 (2024).

[3] Wibisono, Andre, Yihong Wu, and Kaylee Yingxi Yang. "Optimal score estimation via empirical bayes smoothing." arXiv preprint arXiv:2402.07747 (2024).

[4] Zhang, Kaihong, et al. "Minimax Optimality of Score-based Diffusion Models: Beyond the Density Lower Bound Assumptions." arXiv preprint arXiv:2402.15602 (2024).

[5] Oko, Kazusato, Shunta Akiyama, and Taiji Suzuki. "Diffusion models are minimax optimal distribution estimators." International Conference on Machine Learning. PMLR, 2023.

**Questions:**

See weaknesses.

---

> ### Author Rebuttal · Authors · 2024-08-07
>
> Comparison to previous work
> ----
>
> We would like to clarify that most of the prior work in the literature, including the works you mentioned, ask about the *approximation* power of neural networks for representing the score of *arbitrary* distributions, and/or make strong assumptions on the distribution. Typically, for $d$ dimensional distributions, the number of parameters necessary is *exponential in $d$* -- indeed, the bounds shown in all the works you mentioned suffer from this exponential dependence. But in practice, neural networks are surprisingly powerful function approximators of real-world functions, and they seem to be able to represent $d$-dimensional distributions with much less than $\exp(d)$ parameters.
>
> To circumvent this issue, we take a different route: we assume that a $P$ parameter neural network can approximate the score of the distribution that we are trying to learn.   Note that such an assumption is *necessary*: if our network cannot even represent the scores, there is no hope of learning it.
>
> We then ask: if $P$ parameters suffice, how many samples do we need to learn this approximation? We show that the sample complexity for this problem is only *polynomial* in all relevant parameters, and even logarithmic in the Wasserstein error, unlike *all* prior works. We believe that this question and our answer more accurately capture the situation in practice than the one above.
>
> We appreciate the feedback to provide a table -- please see the general response.
>
> Note that [HRX24], which is focused on the two-layer network case in the NTK regime, actually makes the *assumption* that gradient descent in RKHS using a **sufficient** number of samples is sufficient to learn the score (Assumption 3.11). Our paper *proves* that a small number of samples is sufficient to learn the score under the reasonable assumption that the network can represent it; moreover, we provide a quantitative  bound that is polynomial in all relevant parameters.
>
>
>
> Technical contributions
> ---
>
> Here is a brief summary of our technical contributions:
>
> * We exploit the fact that recent works on *sampling* only require an error of $\varepsilon^2/\sigma_t^2$ for each $t$ to show a sample complexity bound *independent* of $\sigma_t$ for score estimation with this error.  That is, the accuracy required fortuitously matches the accuracy achievable at each scale $\sigma_t$, with a sample complexity independent of $\sigma_t$.
>
> * This lets us run until $\sigma_t$ is extremely small; this leads to a  sample complexity *logarithmic* in the final Wasserstein error.
>
> * To show our score estimation result for a single time $t$, instead of making the strong assumption that the distribution is bounded as in [BMR20], we observe that *with high probability* the score at time $t$ is bounded in terms of $\sigma_t$. This observation, combined with a technical argument allows us to remove the $R^3$ dependence in the sample complexity in [BMR20].
>
> * Other contributions that we will refrain from describing in detail here in the interest of space -- we show that we learn the score in a new $1-\delta$ robust sense, rather than standard $L^2$ to circumvent barriers in learning in $L^2$, and we make use of a careful net argument to obtain our *exponential* improvements in the  dependencies on the depth and range of the neural network relative to [BMR20].
>
>
> Section 4
> ---
>
> - As above, section 4 shows that learning in $L^2$ *requires* $\text{poly}(1/\gamma)$ samples to obtain a final Wasserstein error of $\gamma$. We can circumvent this barrier by learning the score in our weaker $1-\delta$ robust sense, to obtain a final sample complexity only logarithmic in $1/\gamma$.
>
> - Minimax optimality: As explained above, prior works show minimax optimality with respect to the class of arbitrary distrbutions, which results in an exponential in $d$ sample complexity. This does not accurately capture the situation in practice -- neural networks are surprisingly powerful approximators of real-world functions and seem to be able to represent $d$ dimensional distributions with much less than $\exp(d)$ parameters/samples. We circumvent this by making the reasonable assumption that the scores of the distribution can be represented with a $P$ parameter neural network.
>
> [HRX24]: Neural Network-Based Score Estimation in Diffusion Models: Optimization and Generalization. Yinbin Han, Meisam Razaviyayn, Renyuan Xu. https://arxiv.org/abs/2401.15604

---

> ### Comment · Reviewer_6Nj2 · 2024-08-12
>
> Thank you for your detailed response. I have raised the score. However, the presentation should still be improved before publication.

---

### Official Review · Reviewer_tUeu · 2024-07-14

**Soundness:** 3
**Presentation:** 3
**Contribution:** 2
**Rating:** 5
**Confidence:** 4

**Summary:**

In this paper, the authors studied the sample complexity of training diffusion models. By using a sufficiently expressive neural network, the authors showed an exponential improvement in the dependence on Wasserstein error and network width, which is expressed as $\tilde{O}(d^2PD\log\Theta \log^3(1/\gamma)/\varepsilon^3)$. This bound has a better polynomial dependence on the dimension $d$ and $P$, a better exponential dependence on $\Theta, \gamma$ and $D$ but a worse polynomial dependence on $\varepsilon$ as a cost.

**Strengths:**

This paper is clearly written and the presentation is fairly good. The theorems and lemmas proposed in this paper are sound and solid.

**Weaknesses:**

This contribution of this paper seems not to be enough for an accept of a top tier machine learning conference. As we know, the TV error or Wasserstein error of learning diffusion models mainly lies in the discretization error caused by solving backward ODE (or SDE) through Euler Maruyama method, as well as the score estimation error. The discretization error is dominated by the latter error according to Chen et al.'s work. In this paper, the authors mainly studied the score estimation error. My biggest concern is that inserting a better generalization bound of score estimation (with regard to polynomial dimensional dependence) is not novel enough. For more questions, please refer to the next section.

**Questions:**

1. For the assumption A2, I think you can directly put the result of Theorem C.3 into the main text since the function approximation is also important, and the statement like "sufficiently small" doesn't seem to be strict.
2. Please provide some explanations on the novelty of the sample complexity theory, other than an insert of polynomial dimension learning error rate of expressive neural networks.
3. The work focuses on DDPM. However, in the original work of Song et al, they used VE (variance exploding) for consistency training. Is it possible for the sample complexity bound in the paper adapted to the VE case?

**Limitations:**

There is not potential societal impact of this work since it is a completely theoretical work.

---

> ### Author Rebuttal · Authors · 2024-08-07
>
> Thank you for your comments and questions.  As you state, prior work
> like Chen et al. have shown that "sampling is as easy as learning the
> score."  Thus the main open question is: how easy *is* learning the
> score?
>
> We think that question is clearly important enough for top tier
> machine learning conferences.  We address it by providing the *first* polynomial sample complexity bound, under the assumption that the score can be effectively represented by neural networks. And this assumption is
> both reasonable (neural networks have proven to be surprisingly
> powerful function approximators) and necessary (if the distribution's
> score isn't well approximated by a neural network, diffusion won't
> work).
>
> **Assumption A2**:
>
> Thank you for the suggestion, which is the consensus of reviewers.  We will include the precise definition of ``sufficiently small" into the main text in the final version of the paper. As stated in C.3., the required $L^2$ error is $\frac{\delta \varepsilon^3}{N^2 \sigma^2
> \log \frac{d}{\gamma}}$.
>
> **Technical novelties**:
>
> Here is a brief summary of our technical contributions:
>
> * We exploit the fact that recent works on *sampling* only require an error of $\varepsilon^2/\sigma_t^2$ for each $t$ to show a sample complexity bound *independent* of $\sigma_t$ for score estimation with this error.  That is, the accuracy required fortuitously matches the accuracy achievable at each scale $\sigma_t$, with a sample complexity independent of $\sigma_t$.
>
> * This lets us run until $\sigma_t$ is extremely small; this leads to a  sample complexity *logarithmic* in the final Wasserstein error.
>
> * To show our score estimation result for a single time $t$, instead of making the strong assumption that the distribution is bounded as in [BMR20], we observe that *with high probability* the score at time $t$ is bounded in terms of $\sigma_t$. This observation, combined with a technical argument allows us to remove the $R^3$ dependence in the sample complexity in [BMR20].
>
>
>
> * Other contributions that we will refrain from describing in detail here in the interest of space -- we learn the score in a new $1-\delta$ robust sense, rather than standard $L^2$, to circumvent barriers in learning in $L^2$, and we make use of a careful net argument to obtain our *exponential* improvements in the  dependencies on the depth and range of the neural network relative to [BMR20].
>
> **VE vs VP:**
>
> The VE and VP processes are simply reparameterizations of each other. That is, if $x_t$ denotes a solution to the VE SDE, then $y_t = e^{-t} x_{e^{2t}-1}$ is a solution to the VP SDE. The reverse process and its discretization can be easily adjusted to match this change of variables.  So, these processes are pretty much identical, and the same results hold.

---

### Author Rebuttal · Authors · 2024-08-07

We thank the reviewers for their feedback.

One common request was for a table placing our results in context of related work.  Here it is:

| **Work** | **Sample Complexity**                                                                                                   | **Notes**                                              |
|-------------|------------------------------------------------------------------------------------------------------------------------------|------------------------------------------------------------------|
| [ZYLL24]    | $\widetilde O\left( \frac{1}{\epsilon^2 \gamma^{d/2}}\right)$                                                                | Assuming Distribution is $\alpha$-subgaussian, satisfies an LSI.  Gaussian Kernel estimator, rather than NN.                       |
| [WWY24]     | $\widetilde O\left(\frac{d^{d/2} \alpha^d R^{d+2}}{\gamma^{d+2} \epsilon^{d+4}} \right)$                                     | distribution is $\alpha$-subgaussian. Gaussian Kernel estimator, rather than NN.                       |
| [OST23]     | $\widetilde O\left( \frac{1}{\epsilon^{O(d)}}\right)$                                                                        | Density supported on $[-1, 1]^d$, belongs to a Besov space       |
| [CHZ+23]    | $\widetilde O\left(\frac{1}{(\epsilon \gamma)^{O(d)}} \right)$                                                               | Assuming density supported on a $d$-dimensional subspace         |
| [BMR20]     | $\widetilde O\left(\frac{d^{5/2}R^3}{\gamma^3 \epsilon^2} \left(\Theta^2 P \right)^D \sqrt{D} \log \frac{1}{\delta} \right)$ | Assuming NN can represent scores, distribution is bounded        |
| Ours        | $\widetilde O\left(\frac{d^2}{\epsilon^3} PD \log \Theta \log^3 \frac{1}{\gamma} \right)$                                    | Assuming NN can represent scores                                 |

Our paper primarily discusses [BMR20] because it is the only related work that is not exponential in the data dimension $d$; but as you can see, it is exponential in the depth $D$ of the neural network.
Ours is the only work giving sample complexity polynomial in these parameters.


[WWY24]: Wibisono, Andre, Yihong Wu, and Kaylee Yingxi Yang. "Optimal score estimation via empirical bayes smoothing." arXiv preprint arXiv:2402.07747 (2024).

[ZYLL24]: Zhang, Kaihong, et al. "Minimax Optimality of Score-based Diffusion Models: Beyond the Density Lower Bound Assumptions." arXiv preprint arXiv:2402.15602 (2024).

[OST23]: Oko, Kazusato, Shunta Akiyama, and Taiji Suzuki. "Diffusion models are minimax optimal distribution estimators." International Conference on Machine Learning. PMLR, 2023.

[CHZ+23]: Chen, Minshuo, et al. "Score approximation, estimation and distribution recovery of diffusion models on low-dimensional data." International Conference on Machine Learning. PMLR, 2023.

[BMR20]: Generative Modeling with Denoising Auto-Encoders and Langevin Sampling. Adam Block, Youssef Mroueh, Alexander Rakhlin. https://arxiv.org/abs/2002.00107

---

### Decision · Program_Chairs · 2024-09-25

**Decision:**

Accept (poster)

**Comment:**

Thank you for your valuable contribution to Neurips and the ML community. Your submitted paper has undergone a rigorous review process, and I have carefully read and considered the feedback provided by the reviewers.

The paper analyzes the sample complexity of training a score-based diffusion model. The sample complexity is improved with respect to certain parameters by avoiding an L^2 accuracy on the score function as done in the previous work. Overall, the paper received mostly positive response from the reviewers. Some issues raised by the reviewers were successfully addressed in the rebuttal. These issues include (i) significance of the main contribution,  (ii) presentation issues (moving Theorem C.3 to the main text) (iii) clearly positioning the result relative to prior-work.

Given this positive assessment, I am willing to recommend the acceptance of your paper for publication.

I would like to remind you to carefully review the reviewer feedback and the resulting discussion. While most reviews were positive, the reviewers have offered valuable suggestions that can further strengthen the quality of the paper. In particular, there were valuable suggestions to reorganize the theorems. Please take another careful look a the 'weaknesses' section of each reviewer comment and the issues (i)-(iii) listed above. I encourage you to use this feedback to make any necessary improvements and refinements before submitting the final version of your paper.

Once again, thank you for submitting your work to Neurips.

Best,